# Towards Syn-to-Real IQA: A Novel Perspective on Reshaping Synthetic Data Distributions

**Aobo Li**    **Jinjian Wu**[*]   **Yongxu Liu**    **Leida Li**    **Weisheng Dong**
Xidian University
abli@stu.xidian.edu.cn, {jinjian.wu, wsdong}@mail.xidian.edu.cn,
{yongxu.liu, ldli}@xidian.edu.cn

## Abstract

Blind Image Quality Assessment (BIQA) has advanced significantly through deep learning, but the scarcity of large-scale labeled datasets remains a challenge. While synthetic data offers a promising solution, models trained on existing synthetic datasets often show limited generalization ability. In this work, we make a key observation that representations learned from synthetic datasets often exhibit a discrete and clustered pattern that hinders regression performance: features of high-quality images cluster around reference images, while those of low-quality images cluster based on distortion types. Our analysis reveals that this issue stems from the distribution of synthetic data rather than model architecture. Consequently, we introduce a novel framework SynDR-IQA, which reshapes synthetic data distribution to enhance BIQA generalization. Based on theoretical derivations of sample diversity and redundancy's impact on generalization error, SynDR-IQA employs two strategies: distribution-aware diverse content upsampling, which enhances visual diversity while preserving content distribution, and density-aware redundant cluster downsampling, which balances samples by reducing the density of densely clustered areas. Extensive experiments across three cross-dataset settings (synthetic-to-authentic, synthetic-to-algorithmic, and synthetic-to-synthetic) demonstrate the effectiveness of our method. The code is available at `https://github.com/Li-aobo/SynDR-IQA`.

## 1 Introduction

Blind Image Quality Assessment (BIQA) aims to automatically and accurately evaluate image quality without relying on reference images [1, 2]. It plays a crucial role in enhancing user experience in multimedia applications, improving the robustness of downstream image processing algorithms, and guiding the optimization of image enhancement methods. However, the BIQA task is challenging due to its complexity and high association with human perception.

In recent years, mainstream BIQA methods have greatly surpassed traditional methods due to the powerful representational capabilities of deep learning models. However, the success of deep learning largely relies on large-scale annotated datasets. The high cost of acquiring subjective quality labels limits the growth of existing datasets. The availability of reference images and the controllability of quality degradation in synthetic distortions suggest that low-cost data augmentation through artificially synthesized data appears to be a feasible solution. In practice, training directly on existing synthetic distortion datasets results in suboptimal quality representations with limited generalization capabilities.

We observe a key phenomenon: models trained on synthetic data tend to produce a discrete and clustered feature distribution. Specifically, as shown in Fig. 1, high-quality image features form

---

[*]Corresponding author.

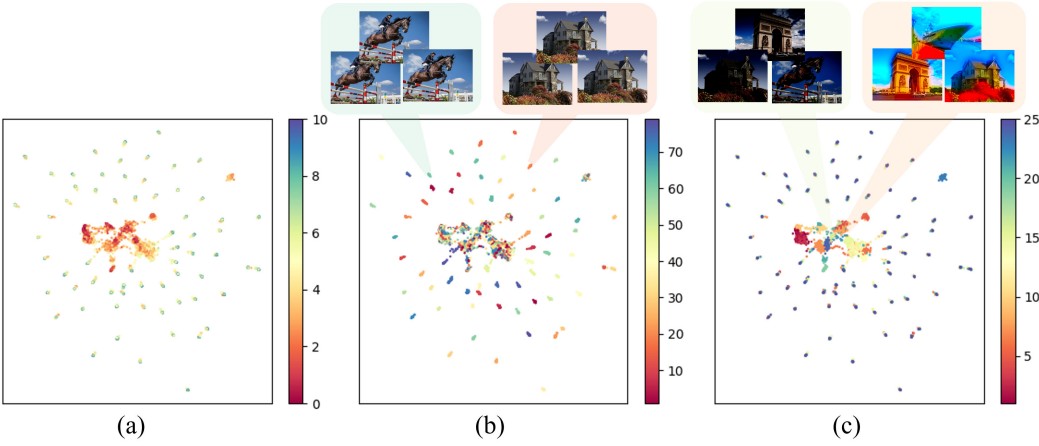

Figure 1: (a), (b), and (c) present UMAP [3] visualizations of the same representations learned from the synthetic distortion dataset KADID-10k [4] by the baseline model [5]. The three visualizations differ only in color mapping: (a) colors indicate the Mean Opinion Score (MOS) values (higher indicates better quality); (b) colors correspond to reference images; and (c) colors denote distortion types. Representative samples are added in (b) and (c) to illustrate the redundancy within high-quality and low-quality clusters, respectively. More visualizations across different backbones and datasets are provided in Appendix A to demonstrate the generality of this phenomenon.

distinct clusters based on reference images, while low-quality image features gather based on distortion types. Medium-quality image features lack smooth transitions and tend to attach to high/low-quality clusters. This discontinuous representation is detrimental to the performance of regression tasks like BIQA [6, 7]. We believe that this phenomenon is primarily caused by two core problems in synthetic distortion datasets:

- 1) **Insufficient content diversity**, which is caused by the limited reference images in synthetic distortion datasets. This leads to the model's tendency to overfit, hindering the formation of a globally consistent quality representation.

- 2) **Excessive redundant samples**, which stem from the distorted images in synthetic distortion datasets being uniform combinations of reference images, distortion types, and distortion intensities. This induces the model to overly focus on these repetitive patterns while neglecting broader information, thereby exacerbating the overfitting problem.

To understand these issues thoroughly, we theoretically derive the impact of sample diversity and redundancy on generalization error. Based on this theoretical foundation, we design a framework called SynDR-IQA from a novel perspective, which reshapes the synthetic data distribution to improve the generalization ability of BIQA. Specifically, **to address the issue of insufficient content diversity**, we propose a Distribution-aware Diverse Content Upsampling (DDCUp) strategy. By sampling reference images from an unlabeled candidate reference set based on the content distribution of existing training set to generate distorted images, we increase the diversity of visual instances, helping the model learn consistent representations across different content. To label the newly generated distorted images, we employ a key assumption: similar content under the same distortion conditions should result in similar quality degradation. Based on this assumption, we generate pseudo-labels for the newly generated images referencing given labeled data corresponding to similar reference images in the training set, ensuring the reasonableness and consistency of the generated pseudo-labels. **To address the issue of excessive redundant samples**, we design a Density-aware Redundant Cluster Downsampling (DRCDown) strategy. It identifies high-density redundant clusters in the training dataset and selectively removes samples from these clusters while retaining samples from low-density regions. This mitigates the negative impact of redundant samples while alleviating data distribution imbalance, thus helping the model learn more generalizable representations. Our contributions can be summarized as follows:

- We observed the key phenomenon that models trained on synthetic data exhibit discrete and clustered feature distributions, and provide an in-depth analysis of the underlying causes. Through theoretical derivation, we demonstrate the impact of sample diversity and redundancy on the model's generalization error.

- From a novel perspective of reshaping synthetic data distribution, we proposed the SynDR-IQA framework, which includes a DDCUp strategy and a DRCDown strategy, to enhance the generalization capability of BIQA models.

- Extensive experiments across various cross-dataset settings, including synthetic-to-authentic, synthetic-to-synthetic, and synthetic-to-algorithmic, validated the effectiveness of the SynDR-IQA framework. Additionally, as a data-based approach, SynDR-IQA can be integrated with existing model-based methods without adding inference costs.

## 2 Related Work

**Deep Learning-based BIQA Methods.** Deep learning has revolutionized BIQA, leading to significant advancements in accuracy and robustness [8, 9]. Recent works have explored various innovative approaches to address the challenges in this field. Zhu et al. [10] proposed MetaIQA, employing meta-learning to enhance generalization across diverse distortion types. Su et al. [11] introduced a self-adaptive hyper network architecture for adaptive quality estimation in real-world scenarios. Ke et al. [12] developed MUSIQ, a multi-scale image quality transformer processing native resolution images with varying sizes. Saha et al. [13] introduced Re-IQA, an unsupervised mixture-of-experts approach that jointly learns complementary content and quality features for perceptual quality prediction. Shin et al. [14] proposed QCN, utilizing comparison transformers and score pivots for improving cross-dataset generalization. Xu et al. [15] demonstrated the effectiveness of injecting local distortion features into large pretrained vision transformers for IQA tasks. Despite these advancements, the success of deep learning-based BIQA methods heavily relies on large-scale annotated datasets. The high cost and time-consuming nature of acquiring subjective quality labels for real-world images significantly limit the growth of existing datasets. This limitation has prompted researchers to explore the potential of leveraging synthetic distortions to generalize to real-world distortions.

**Synthetic-to-Real Generalization in BIQA.** Due to the significant domain differences between synthetic distortions and real-world distortions, models trained on synthetic distortion data often perform poorly when facing real-world images. To bridge this gap, several studies have explored Unsupervised Domain Adaptation (UDA) techniques. Chen et al. [16] proposed a curriculum-style UDA approach for video quality assessment, adapting models from source to target domains progressively. Lu et al. [17] introduced StyleAM, aligning source and target domains in the feature style space, which is more closely associated with image quality. Li et al. [18] developed FreqAlign, which excavates perception-oriented transferability from a frequency perspective, selecting optimal frequency components for alignment. Most recently, Li et al. [19] proposed DGQA, a distortion-guided UDA framework that leverages adaptive multi-domain selection to match data distributions between source and target domains, reducing negative transfer from outlier source domains. However, previous work has neglected the distributional issues of synthetic distortion datasets. In this work, we introduce a novel framework SynDR-IQA to enhance the syn-to-real generalization ability of BIQA by reshaping the distribution of synthetic data.

## 3 Methodology

In this section, we introduce the SynDR-IQA framework, which aims to enhance the generalization capability of BIQA models by reshaping the synthetic data distribution. We begin with a problem formalization for BIQA, establishing the foundational context for our work. Then, we conduct a theoretical analysis exploring the impact of sample diversity and redundancy on the generalization error. Building upon this theoretical foundation, we detail the two core components of SynDR-IQA: Distribution-aware Diverse Content Upsampling (DDCUp), which addresses the challenge of limited diversity in synthetic datasets and Density-aware Redundant Cluster Downsampling (DRCDown), which mitigates the issue of data redundancy.

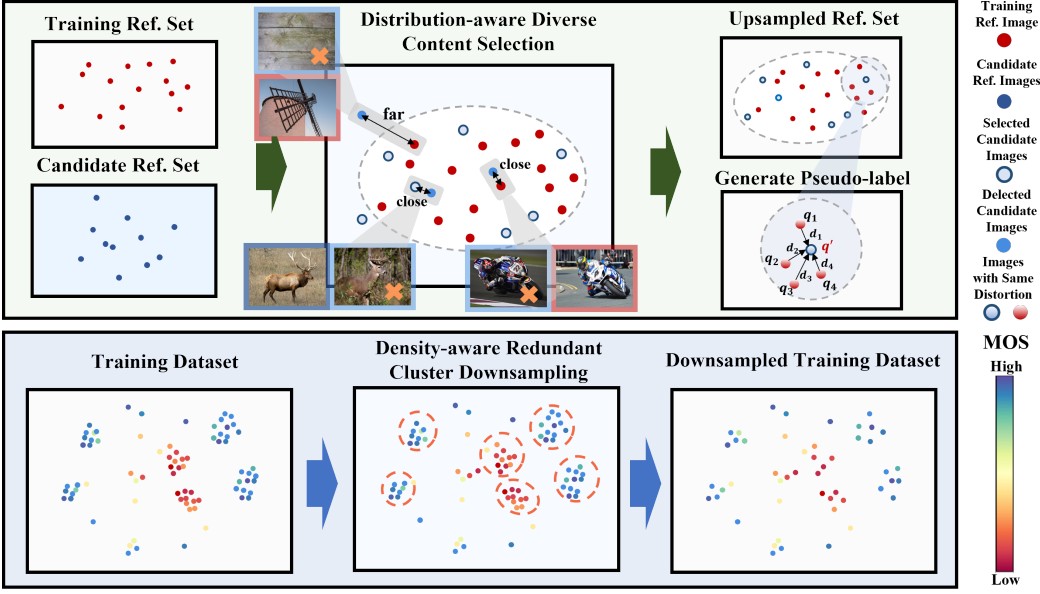

Figure 2: This figure shows the core concepts of two strategies in SynDR-IQA. The DDCUp strategy (upper) selects images from candidate reference sets that are similar in distribution but diverse in content to the training reference sets for synthesizing distorted samples. The pseudo-labels of these samples depend on the nearest neighbors of their reference images. The DRCDown strategy (lower) identifies high-density clusters in the training dataset and selectively removes samples from these clusters while retaining samples from low-density regions.

## 3.1 Problem Formalization

BIQA aims to predict the perceptual quality of images without reference. Let $\mathcal{X}$ denote the space of all possible images, and $\mathcal{Y} = [0, 1]$ represent the range of quality scores (for simplicity). The BIQA task is formalized as learning a function $f : \mathcal{X} \rightarrow \mathcal{Y}$ that maps an input image to its quality score. The optimal function $f^*$ is defined by minimizing the expected risk: $f^* = \arg\min_{f \in \mathcal{F}} \mathbb{E}_{(x,y) \sim \mathcal{D}}[\mathcal{L}(f(x), y)]$, where $\mathcal{F}$ is the hypothesis space, $\mathcal{D}$ is the true distribution of images and quality scores, and $\mathcal{L}$ is a loss function. Since the true data distribution is inaccessible, in practice, we instead minimize the empirical error on the training dataset $\hat{\mathcal{D}} = \{(x_i, y_i)\}_{i=1}^n$: $\hat{f} = \arg\min_{f \in \mathcal{F}} \frac{1}{n} \sum_{i=1}^n \mathcal{L}(f(x_i), y_i)$.

## 3.2 Theoretical Analysis

The construction of synthetic distortion datasets follows a systematic process: applying predefined distortion types at various intensity levels to a set of reference images [4]. This generation mechanism exhibits two key characteristics: First, low-intensity distortions produce images nearly identical to their references, while high-intensity distortions generate images predominantly characterized by distortion-specific patterns (representative visual examples can be found in Appendix B). Second, the dataset generation typically employs uniform sampling across reference images, distortion types, and intensity levels. These two characteristics jointly lead to a fundamental issue: samples are drawn from different local distributions rather than following identical sampling from the true distribution, resulting in a discretely clustered structure in the distribution space.

To understand how this clustered distribution affects model generalization, we need to extend the classical generalization error analysis to account for samples being drawn from different local distributions rather than the true underlying distribution. To formalize this extension, we model the synthetic dataset $\hat{\mathcal{D}} = \{(x_i, y_i)\}_{i=1}^n$ as comprising $m$ clusters, where each cluster consists of one *i.i.d.* sample from the true distribution $\mathcal{D}$, along with its associated $k_i - 1$ samples drawn from the corresponding local distribution $\mathcal{D}_i \subset \mathcal{D}$. This formal characterization leads to the following generalization bound:

**Theorem 3.1** (Generalization Bound for Clustered Synthetic Data). *Let $\mathcal{F}$ be a hypothesis class of functions $f : \mathcal{X} \to \mathcal{Y}$, and $\hat{\mathcal{D}} = \{(x_i, y_i)\}_{i=1}^{n}$ be a dataset consisting of $m$ i.i.d. samples from true distribution $\mathcal{D}$, along with their respective $k_i - 1$ redundant samples drawn from the corresponding local distributions $\mathcal{D}_i \subset \mathcal{D}$. With probability at least $1 - \delta$, for all $f \in \mathcal{F}$, we have:*

$$|R(f) - R_{\text{emp}}(f)| \leq 2\,\text{Rad}_m(\mathcal{F}) + \sqrt{\frac{2\log(2/\delta)}{m}} + \sqrt{\frac{\eta\log(2/\delta)}{8m}} + \frac{2\log(2/\delta)}{3m}. \quad (1)$$

*where $R(f)$ is the true risk, $R_{emp}(f)$ is the empirical risk, $\text{Rad}_m(\mathcal{F})$ is the Rademacher complexity, and $\eta = \frac{1}{m}\sum_{i=1}^{m}\frac{1}{k_i}$ is defined as redundancy heterogeneity which quantifies the degree of imbalanced distribution of redundant samples.*

According to Theorem 3.1, we can observe that the upper bound of the generalization error is influenced by the Rademacher complexity $\text{Rad}_m(\mathcal{F})$, number of *i.i.d.* samples $m$ (also referred to as diverse samples), and redundancy heterogeneity $\eta$. Excluding model-related factors (captured by $\text{Rad}_m(\mathcal{F})$), the bound reveals that the number of distinct samples $m$ plays a primary role in determining the upper bound. Enhancing the sampling of diverse samples can effectively lower this upper bound. Balancing samples to reduce redundancy heterogeneity $\eta$ can also effectively decrease the upper bound of the generalization error. However, indiscriminately increasing samples may lead to an increase in redundant samples from local distributions. This, in turn, can cause higher redundancy heterogeneity $\eta$, thereby amplifying the third term and degrading overall generalization performance.

These theoretical insights motivate us to reshape the sample distribution from the perspectives of **increasing content diversity** (increasing $m$) and **balancing sample density** (decreasing $\eta$) to improve the generalization performance of BIQA.

### 3.3 SynDR-IQA Framework

Building upon the theoretical insights, we propose the SynDR-IQA framework as shown in Fig. 2, which consists of two primary strategies: distribution-aware diverse content upsampling and density-aware redundant cluster downsampling. These strategies collectively aim to reshape the synthetic data distribution to obtain more generalizable BIQA models.

#### 3.3.1 Distribution-aware Diverse Content Upsampling

Our DDCUp strategy aims to enrich training data by incorporating diverse visual content while maintaining the original content distribution.

---

**Algorithm 1** Distribution-aware Diverse Content Upsampling Strategy

---

**Require:** Training dataset $\mathcal{D}$, Training reference set $\mathcal{D}_r$, Candidate reference set $\mathcal{D}_c$, Feature extractor $f(\cdot)$, Distance metric $\text{Dist}(\cdot)$
1: Initialize $\mathcal{D}_r^{new} \leftarrow \emptyset$
2: $DistT \leftarrow \{\text{Dist}(f(x_r^i), f(x_r^j)) | x_r^i, x_r^j \in \mathcal{D}_r, i \neq j\}$
3: **for** each $x_c \in \mathcal{D}_c$ **do**
4:  $DistC \leftarrow \text{Dist}(f(x_c), f(\mathcal{D}_r))$
5:  **if** $\text{Min}(DistC) > \text{Median}(DistT)$ and $\text{Max}(DistC) < \text{Max}(DistT)$ **then**
6:   $DistN \leftarrow \text{Dist}(f(x_c), f(\mathcal{D}_r^{new}))$
7:   **if** $\mathcal{D}_r^{new}$ is $\emptyset$ or $\text{Min}(DistN) > \text{Median}(DistT)$ **then**
8:    $\mathcal{D}_r^{new} \leftarrow \mathcal{D}_r^{new} \cup \{x_c\}$
9:   **end if**
10:  **end if**
11: **end for**
12: $\mathcal{D}' \leftarrow \mathcal{D} \cup \text{GenSyn}(\mathcal{D}, \mathcal{D}_r, \mathcal{D}_r^{new}, ...)$
13: **return** Upsampled dataset $\mathcal{D}'$

---

Algorithm 1 details the DDCUp strategy. We select additional reference images from KADIS-700k [4] to build the candidate reference set $\mathcal{D}_c$. To avoid introducing excessive noise, we limit its size

to be equal to that of the training reference set. A feature extractor $f(\cdot)$, pre-trained on ImageNet, is used to extract features for reference images. A distance metric, $\text{Dist}(\cdot)$ (cosine distance in our implementation), is utilized to guide the selection process. Our algorithm ensures that the selected images do not deviate excessively from the original content distribution while also being sufficiently distinct from each other (lines 5 and 7), thereby promoting diversity and preventing redundancy. A qualitative analysis in Appendix D visually demonstrates these effects.

---

**Algorithm 2** Synthetic Data Generation

---

**Require:** Training dataset $\mathcal{D}$, Training reference set $\mathcal{D}_r$, New reference set $\mathcal{D}_r^{new}$, Feature extractor $f(\cdot)$, Distance metric $\text{Dist}(\cdot)$, Number of nearest neighbors $k$, Reference feature distance threshold $T_{rf}$, Distortion algorithm $\text{GenDist}(\cdot)$, Number of distortion types $T$, Number of distortion level $L$

1: Initialize $\mathcal{D}_{GenSyn} \leftarrow \emptyset$
2: **for** each $x_{new} \in \mathcal{D}_r^{new}$ **do**
3:     $nns \leftarrow \text{kNN}(f(x_{new}), f(\mathcal{D}_r), k, \text{Dist})$ {k-Nearest Neighbors of $x_{new}$}
4:     $nns \leftarrow \{nn \mid \text{Dist}(f(x_r^{nns[0]}), f(x_{new})) - \text{Dist}(f(x_r^{nn}), f(x_{new})) < T_{rf}, nn \in nns\}$
5:     $w \leftarrow \text{Softmax}(\{\text{Dist}(f(x_r^{nn}), f(x_{new})) \mid nn \in nns\})$
6:     **for** each $t \in T, l \in L$ **do**
7:         $x_{new}^{(t,l)} \leftarrow \text{GenDist}(x_{new}, t, l)$, $y_{new}^{(t,l)} \leftarrow \sum_{nn \in nns} w_{nn} y_{nn}^{(t,l)}$
8:         $\mathcal{D}_{GenSyn} \leftarrow \mathcal{D}_{GenSyn} \cup \{(x_{new}^{(t,l)}, y_{new}^{(t,l)})\}$
9:     **end for**
10: **end for**
11: **return** Generated synthetic dataset $\mathcal{D}_{GenSyn}$

---

After selecting diverse reference images, we generate corresponding distorted images and pseudo-labels to augment the training dataset. Algorithm 2 details this process. For each selected reference image, we apply the same distortion generation process used to create the training dataset. Notably, to reduce the increase of redundant samples and prevent additional label noise, only distortion intensities of levels 1, 3, and 5 are implemented. To generate reliable pseudo-labels for these newly distorted images, we leverage the assumption that similar content under the same distortion conditions should result in similar quality degradation. For each new reference image $x_{new}$, we identify its k nearest neighbors (kNN) within the original training set's reference images based on the feature distances. To further enhance the reliability of the pseudo-labels, we filter these nearest neighbors, retaining only those whose feature distance to $x_{new}$ is within a certain threshold $T_{rf}$ (set to 0.05) (line 4). The pseudo-label for each distorted image of $x_{new}$ is then calculated as a weighted average of the labels of its nearest neighbors' corresponding distorted images, where the weights are determined by the softmax of the feature distances.

### 3.3.2 Density-aware Redundant Cluster Downsampling

An overabundance of redundant samples can bias the model, hindering its ability to generalize to unseen data. It also contributes to increased redundancy heterogeneity ($\eta$ in Theorem 3.1), further increasing the generalization error. Therefore, we propose a DRCDown strategy to mitigate the negative impact of redundant samples and reduce $\eta$, thereby further enhancing the model's generalization performance.

Algorithm 3 details the DRCDown strategy. Different from DDCUp, it obtains features before each training round, to ensure distortion-level redundancy reduction while making efficient use of training data. We identify pairs of similar samples based on both the distance metric $\text{Dist}(\cdot)$ and label distance (L1 in our implementation) (line 3). It ensures that we remove redundancy without discarding hard samples with tiny feature difference yet large quality difference. The distorted-feature distances threshold $T_{df}$ and the ground-truth distances threshold $T_g$ are set to 0.1, and 1 (for MOS values in $[0, 10]$), respectively. By considering both feature and label similarity, we aim to specifically target and reduce the density of high-density clusters that contribute significantly to redundancy heterogeneity.

After identifying similar sample pairs, we employ a disjoint set union (DSU) data structure to group these pairs into clusters (line 7). For each cluster whose size is greater than $2T_u$, we randomly downsample it to $\max(\lfloor N_u/2 \rfloor, T_u)$ samples, where $N_u$ is the original cluster size (line 10). This

**Algorithm 3** Density-aware Redundant Cluster Downsampling Strategy

---

**Require:** Training dataset $\mathcal{D}$, Feature extractor $f(\cdot)$, Distance metric $Dist(\cdot)$, Distorted-feature distances threshold $T_{df}$, Ground-truth distances threshold $T_g$, Threshold for minimum union size $T_u$

1: Initialize $\mathcal{D}_{down} \leftarrow \emptyset, SimPairs \leftarrow \emptyset$
2: **for** each $(x_i, y_i), (x_j, y_j) \in \mathcal{D}$ **do**
3:     **if** $Dist(f(x_i), f(x_j)) < T_{df}$ **and** $|y_i - y_j| < T_g$ **then**
4:         $SimPairs \leftarrow SimPairs \cup \{(x_i, x_j)\}$
5:     **end if**
6: **end for**
7: $Unions \leftarrow DSU(SimPairs)$ {Union disjoint sets of similar pairs}
8: **for** each $u \in Unions$ **do**
9:     **if** $Length(u)/2 > T_u$ **then**
10:         $u \leftarrow \{$randomly select $Max(\lfloor Length(u)/2 \rfloor, T_u)$samples among union$\}$
11:     **end if**
12:     $\mathcal{D}_{down} \leftarrow \mathcal{D}_{down} \cup u$
13: **end for**
14: **return** Downsampled dataset $\mathcal{D}_{down}$

---

threshold $T_u$ prevents excessive downsampling, ensuring that the downsampled dataset retains sufficient information for effective training. By selectively removing samples from over-represented regions, the DRCDown strategy effectively reduces redundancy and promotes a more balanced data distribution, directly addressing the issue of high redundancy heterogeneity and thereby contributing to improved generalization performance. This reduction in $\eta$ helps to lower the generalization error bound as established in Theorem 3.1, leading to a more robust and generalizable IQA model.

## 4 Experiments

### 4.1 Experimental Setups

**Datasets and Protocols.** We conduct experiments on eight IQA datasets: four synthetic distortion datasets LIVE [20], CSIQ [21], TID2013 [22], KADID-10k [4], three authentic distortion datasets LIVEC [23], KonIQ-10k [24], BID [25], and the dataset PIPAL [26] with both synthetic and algorithmic distortions. Model performance is assessed using Spearman's Rank Correlation Coefficient (SRCC) and Pearson's Linear Correlation Coefficient (PLCC). Both coefficients range from -1 to 1, with values closer to 1 indicating better performance.

**Implementation Details.** In our experiments, we use the same model architecture (ResNet-50 [5]) and loss function (L1Loss) as DGQA [19]. For synthetic-to-authentic and synthetic-to-algorithmic settings, models are trained using distortion types selected by DGQA, while for the synthetic-to-synthetic setting, all distortion types are used. Following standard IQA protocols [27, 28], we employed an 80/20 split by reference images for intra-dataset experiments, repeated ten times with median SRCC/PLCC reported. For cross-database experiments, models are trained on KADID-10k and tested on other datasets. During training, one $224 \times 224$ patch is randomly sampled from each image, with random horizontal flipping applied for data augmentation. The mini-batch size is set to 32, with a learning rate of $2 \times 10^{-5}$. The Adam optimizer, with a weight decay of $5 \times 10^{-4}$, is used to optimize the model for 24 epochs. During testing, the predictions from five patches per image are averaged for the final output. All experiments are implemented in PyTorch and on a server equipped with a 2.10GHz Intel Xeon(R) CPU E5-2620 v4 processor and four NVIDIA GTX 1080 Ti GPUs.

### 4.2 Performance Evaluation

**Performance on the Synthetic-to-Authentic Setting.** We first evaluate the generalization capability of SynDR-IQA when transferring from synthetic to authentic distortions. The results of the comparison between SynDR-IQA and state-of-the-art methods are summarized in Table 1. SynDR-IQA achieves leading performance across all three authentic datasets, with only slight underperformance against FreqAlign (KADID-10k→KonIQ-10k, SRCC) and Q-Align (KADID-10k→LIVEC, PLCC). Compared to the next best method, DGQA, our method improves the average SRCC and PLCC

Table 1: Performance comparison on the synthetic-to-authentic setting (KADID-10k→LIVEC, KonIQ-10k, and BID).

| Methods | LIVEC | | KonIQ-10k | | BID | | Average | |
|---|---|---|---|---|---|---|---|---|
| | SRCC | PLCC | SRCC | PLCC | SRCC | PLCC | SRCC | PLCC |
| RankIQA [29] | 0.491 | 0.495 | 0.603 | 0.551 | 0.510 | 0.367 | 0.535 | 0.471 |
| DBCNN [30] | 0.572 | 0.589 | 0.639 | 0.618 | 0.620 | 0.609 | 0.613 | 0.606 |
| HyperIQA [11] | 0.490 | 0.487 | 0.545 | 0.556 | 0.379 | 0.282 | 0.472 | 0.442 |
| MUSIQ [12] | 0.517 | 0.524 | 0.554 | 0.573 | 0.575 | 0.600 | 0.549 | 0.566 |
| VCRNet [31] | 0.561 | 0.548 | 0.517 | 0.525 | 0.542 | 0.545 | 0.540 | 0.540 |
| KGANet [32] | 0.575 | - | 0.528 | | - | - | - | - |
| CLIPIQA+ [33] | 0.512 | 0.543 | 0.511 | 0.515 | 0.474 | 0.442 | 0.499 | 0.500 |
| Q-Align [34] | 0.702 | **0.744** | 0.668 | 0.665 | - | - | - | - |
| DANN [35] | 0.499 | 0.484 | 0.638 | 0.636 | 0.586 | 0.510 | 0.574 | 0.543 |
| UCDA [16] | 0.382 | 0.358 | 0.496 | 0.501 | 0.348 | 0.391 | 0.408 | 0.417 |
| RankDA [36] | 0.451 | 0.455 | 0.638 | 0.623 | 0.535 | 0.582 | 0.542 | 0.553 |
| StyleAM [17] | 0.584 | 0.561 | 0.700 | 0.673 | 0.637 | 0.567 | 0.640 | 0.600 |
| FreqAlign [18] | 0.618 | 0.588 | **0.748** | 0.721 | 0.674 | 0.708 | 0.680 | 0.673 |
| DGQA [19] | 0.696 | 0.690 | 0.681 | 0.687 | 0.770 | 0.753 | 0.716 | 0.710 |
| SynDR-IQA | **0.713** | 0.714 | 0.727 | **0.735** | **0.788** | **0.764** | **0.743** | **0.737** |

Table 2: Performance comparison on the setting of synthetic-to-algorithmic (KADID-10k→algorithmic distortions on PIPAL).

| Distortion Type | DGQA | | SynDR-IQA | |
|---|---|---|---|---|
| | SRCC | PLCC | SRCC | PLCC |
| Traditional SR | 0.4897 | 0.4567 | $\mathbf{0.5247}_{+3.50\%}$ | $\mathbf{0.4808}_{+2.41\%}$ |
| PSNR-originated SR | 0.5419 | 0.5404 | $\mathbf{0.5845}_{+4.26\%}$ | $\mathbf{0.5680}_{+2.76\%}$ |
| SR with kernel mismatch | 0.5810 | 0.5956 | $\mathbf{0.6263}_{+4.53\%}$ | $\mathbf{0.6342}_{+3.86\%}$ |
| GAN-based SR | 0.1629 | 0.1353 | $\mathbf{0.1998}_{+3.69\%}$ | $\mathbf{0.1629}_{+2.76\%}$ |
| Denoising | 0.5393 | 0.5279 | $\mathbf{0.5749}_{+3.56\%}$ | $\mathbf{0.5552}_{+2.73\%}$ |
| SR and Denoising Joint | **0.5588** | **0.5193** | $0.5557_{-0.31\%}$ | $0.5023_{-1.70\%}$ |
| Average | 0.4789 | 0.4625 | $\mathbf{0.5110}_{+3.21\%}$ | $\mathbf{0.4839}_{+2.14\%}$ |

across the three datasets by 2.7% and 2.7%, respectively. The significant improvement demonstrates the superior syn-to-real generalization capability of SynDR-IQA.

**Performance on the Synthetic-to-Algorithmic Setting.** We further evaluate SynDR-IQA on the synthetic-to-algorithmic setting. Table 2 compares our method with DGQA across different types of algorithmic distortions. SynDR-IQA consistently improves over DGQA for most distortion types. Notably, it outperforms over 4% SRCC for PSNR-originated SR and SR with kernel mismatch. A slight decrease is observed only for SR and Denoising Joint. Overall, these results indicates that SynDR-IQA effectively generalizes to algorithmic distortions, demonstrating its robustness in handling complex and unseen distortion types.

**Performance on the Synthetic-to-Synthetic Setting.** We also evaluate SynDR-IQA on the synthetic-to-synthetic setting. SRCC results are summarized in Table 3. SynDR-IQA shows superior performance over the baseline in both in-dataset and cross-dataset evaluations. On KADID-10k, our method achieves an SRCC of 0.8922, indicating better fitting to the training data. In cross-dataset testing, SynDR-IQA maintains higher performance, demonstrating enhanced generalization to other synthetic distortion datasets. These results confirm that our approach enables the model to learn more robust and generalized feature representations.

## 4.3 Ablation Study

To understand the contributions of each component in SynDR-IQA, we perform ablation experiments on the synthetic-to-authentic setting (SRCC reported in Table 4). Here, *CD* refers to adding the full candidate dataset during training, *CD+SEL* denotes DDCUp, and *DOWN* signifies DRCDown.

Table 3: Performance comparison on the synthetic-to-synthetic setting (single database evaluation on KADID-10k, KADID-10k→LIVE, CSIQ, and TID2013).

| Methods | KADID-10k | | LIVE | | CSIQ | | TID2013 | |
|---|---|---|---|---|---|---|---|---|
| | SRCC | PLCC | SRCC | PLCC | SRCC | PLCC | SRCC | PLCC |
| Baseline | 0.8528 | 0.8526 | 0.9173 | 0.8988 | 0.7965 | 0.8017 | 0.7077 | 0.7220 |
| SynDR-IQA | **0.8922** | **0.8974** | **0.9258** | **0.9014** | **0.8069** | **0.8092** | **0.7147** | **0.7328** |

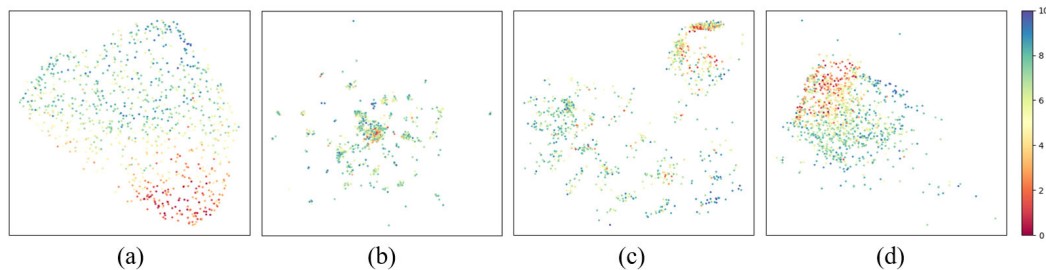

|   (a)   |   (b)   |   (c)   |   (d)   |

Figure 3: UMAP visualization of features extracted from LIVEC using the same model under four different training processes: (a) trained directly on LIVEC, (b) trained directly on KADID-10k), (c) trained on KADID-10k based on DGQA, and (d) trained on KADID-10k based on SynDR-IQA.

**Effect of Candidate Dataset.** Comparing **a)** and **b)**, performance on KonIQ-10k improves, but slightly decreases on LIVEC and BID. This suggests that indiscriminately increasing data to enrich content diversity is not necessarily beneficial, as this may also include samples with significant distributional differences from the training dataset and new redundant samples, thereby hindering model generalization.

**Effect of DDCUp.** Comparing **a)**, **b)**, and **e)**, DDCUp significantly improves performance. By selectively increasing diversity while maintaining data distribution, DDCUp effectively balances content diversity and distribution consistency, leading to better generalization across different datasets.

**Effect of DRCDown.** Comparing **a)** and **c)**, implementing DRCDown alone already shows a consistent improvement over the baseline. This indicates that controlling sample density addresses data redundancy and imbalance, yielding more precise and generalizable feature representations.

The combination of all components (SynDR-IQA) yields the best overall performance, achieving a 2.71% average SRCC improvement over the baseline. These results demonstrate that each proposed component in SynDR-IQA contributes to a more robust and generalized model.

### 4.4 Visualization Analysis

We implement the same model under four different training processes: (a) directly on LIVEC, (b) directly on KADID-10k, (c) on KADID-10k based on DGQA, and (d) on KADID-10k based on SynDR-IQA. We then extract features from LIVEC using these models for UMAP visualization, as shown in Fig. 3. The visualization clearly shows that representations from the model trained directly on KADID-10k form distinct and scattered clusters, indicating poor generalization to authentic distortions. The representations obtained from DGQA show some improvement but remain relatively dispersed. In contrast, SynDR-IQA produces continuous and smooth feature patterns that are much closer to those obtained from the model trained directly on LIVEC. This visually validates SynDR-IQA's effectiveness in bridging the synthetic-to-authentic domain gap.

## 5 Conclusion

In this work, we aim to address the critical challenge of limited generalization ability in BIQA models trained on synthetic datasets. Our investigation reveals a key pattern: representations learned from synthetic datasets tend to form discrete and clustered distributions, with high-quality image features

Table 4: Ablation study of SynDR-IQA components on the synthetic-to-authentic setting. The symbols ✓ and indicate the inclusion of a component.

| Index | CD | SEL | DOWN | LIVEC | KonIQ-10k | BID | Average |
|---|---|---|---|---|---|---|---|
| **a)** | | | | 0.6958 | 0.6810 | 0.7696 | 0.7155 |
| **b)** | ✓ | | | 0.6901 | 0.7105 | 0.7677 | 0.7228 |
| **c)** | | | ✓ | 0.6962 | 0.6887 | 0.7793 | 0.7214 |
| **d)** | ✓ | | ✓ | 0.7095 | 0.7107 | 0.7775 | 0.7326 |
| **e)** | ✓ | ✓ | | **0.7219** | 0.7119 | 0.7822 | 0.7387 |
| **f)** | ✓ | ✓ | ✓ | 0.7127 | **0.7268** | **0.7884** | **0.7426** |

clustering around reference images and low-quality features clustering based on distortion types, which significantly hinders generalization to authentic or unseen distortions. Motivated by this, we theoretically analyze the impact of sample diversity and redundancy on generalization error. Our theoretical insights underpin SynDR-IQA, a novel framework designed to reshape synthetic data distributions to improve BIQA model generalization. SynDR-IQA employs two key strategies: 1) DDCUp, which enhances content diversity while preserving the content distribution of the training dataset; 2) DRCDown, which optimizes sample distribution by reducing the density of dense clusters. Comprehensive experiments across three cross-dataset settings consistently demonstrate that models trained with our SynDR-IQA framework achieve improved generalization ability.

## Acknowledgments and Disclosure of Funding

This work was partially supported by the National Key Research and Development Program of China (2023YFA1008500), the National Natural Science Foundation of China (62401420), and the Hangzhou Joint Fund of the Zhejiang Provincial Natural Science Foundation of China (LHZSZ25F010003).

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

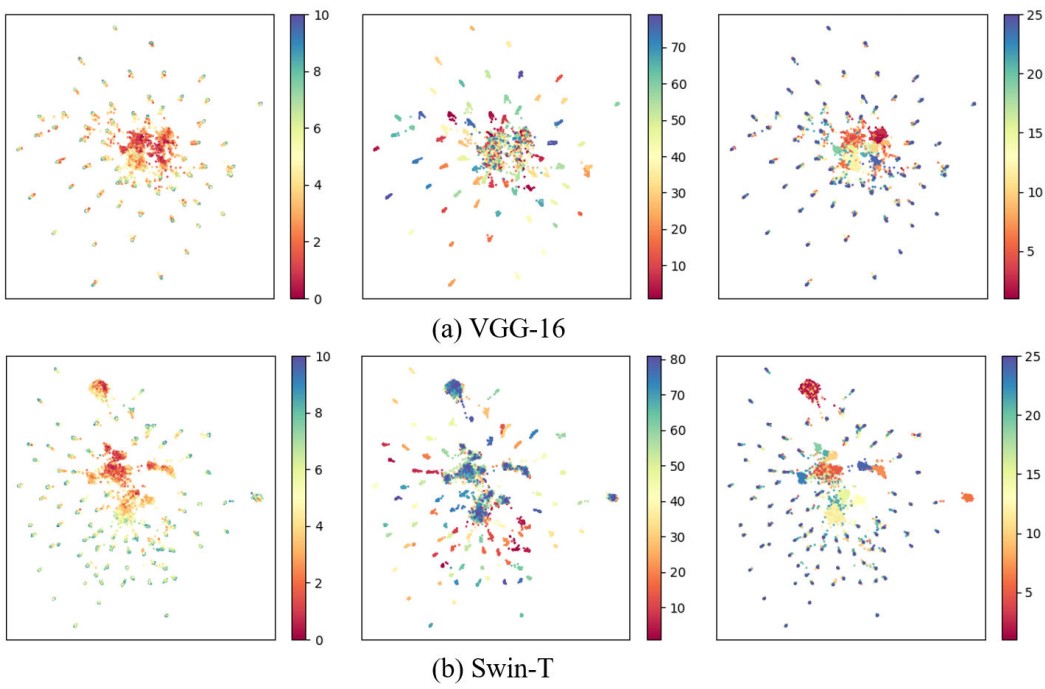

(a) VGG-16

(b) Swin-T

Figure 4: UMAP [3] visualizations of features learned by VGG-16 [37] and Swin-T [38] on KADID-10k [4]. Both show discrete clustering patterns on synthetic distortion data.

## A    UMAP Visualizations across Different Backbones and Datasets

To further verify the generality of the phenomenon observed in Fig. 1 and discussed in the Introduction, we visualize the feature distributions learned on different backbones and datasets. Fig. 4-6 present the UMAP [3] projections of the learned features under various experimental conditions.

Fig. 4 illustrates the results of VGG-16 [37] and Swin Transformer Tiny (Swin-T) [38] trained on KADID-10k [4]. Both models exhibit discrete and clustered feature structures on synthetic distortions: high-quality samples cluster by reference images, and low-quality samples cluster by distortion types. Fig. 5 shows a similar pattern for ResNet-50 [5] trained on TID2013 [22], confirming that this phenomenon is consistent across synthetic datasets. In contrast, Fig. 6 presents the UMAP results of ResNet-50 [5] trained on authentic IQA datasets (LIVEC [23], BID [25], and KonIQ-10k [24]), where features are more continuous and smoothly distributed.

These differences stem from the data distribution itself. Synthetic distortion datasets, constrained by limited content diversity and redundant distortion combinations, tend to drive models toward over-clustered and discontinuous feature spaces. Authentic datasets possess a more balanced and inherently diverse distribution, enabling models to learn more continuous, semantically coherent, and generalizable representations.

## B    Visual Examples of KADID-10k

To further illustrate the distributional characteristics discussed in the main text, we present several representative visual examples from the KADID-10k dataset [4].

Fig. 7 shows part of the low-distortion images, all derived from the same reference image (I01.png). These images exhibit extremely high visual similarity and are almost indistinguishable from their original reference. This suggests that low-intensity distortions have minimal impact on the visual appearance of the images, leading to a high degree of redundancy among samples.

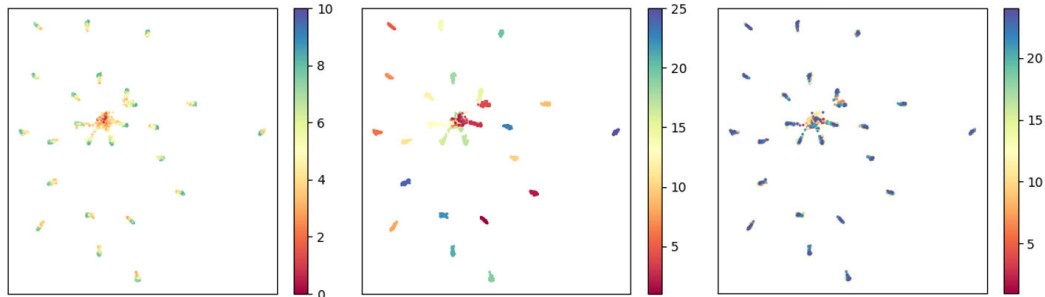

Figure 5: UMAP [3] visualization of ResNet-50 [5] trained on TID2013 [22], demonstrating consistent clustering trends across synthetic datasets.

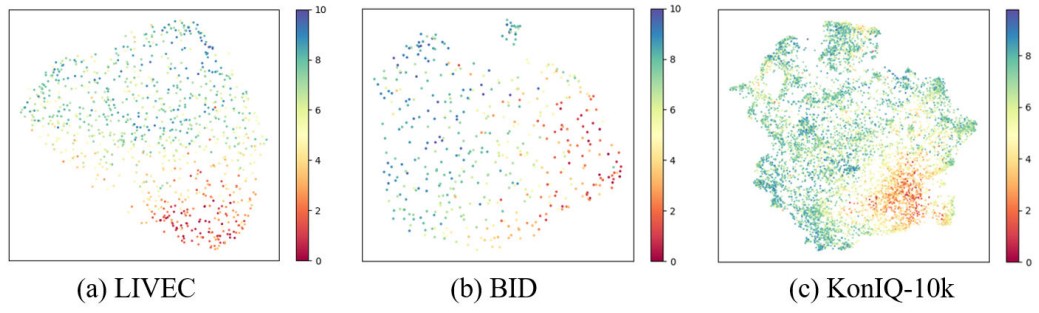

(a) LIVEC       (b) BID       (c) KonIQ-10k

Figure 6: UMAP [3] visualizations of ResNet-50 [5] trained on authentic datasets (LIVEC [23], BID [25], and KonIQ-10k [24]). Features are more continuous and well-distributed, reflecting a more balanced and diverse data structure.

Fig. 8 presents part of the high-distortion images from KADID-10k. The images are dominated by strong and repetitive distortion-specific patterns. It can be observed that examples with the same distortion type show nearly identical artifact structures, and in some cases, even different distortion types may result in similar overall visual patterns. These patterns reveal the strong clustering effect caused by the synthetic distortion process.

These examples visually reinforce the observations discussed in the main text: low-distortion samples are visually redundant, while high-distortion samples often share highly similar, distortion-dependent artifacts. This further demonstrates the discrete and clustered characteristics of the data distribution in synthetic distortion datasets.

## C   The Proof of Theorem 3.1

*Proof.* The key challenge in analyzing clustered data is that samples violate the *i.i.d.* assumption fundamental to standard generalization theory. To address this challenge, We begin by applying the triangle inequality to separate the generalization error into two components:

$$|R(f) - R_{\text{emp}}(f)| \leq |R(f) - R_{\text{m-iid}}(f)| + |R_{\text{m-iid}}(f) - R_{\text{emp}}(f)|, \tag{2}$$

where $R_{\text{m-iid}}(f)$ represents the empirical risk based on $m$ *i.i.d.* samples drawn from the true distribution $\mathcal{D}$. We now bound each term. This decomposition isolates: 1) The standard i.i.d. generalization error $|R(f) - R_{\text{m-iid}}(f)|$, 2) The clustering bias $|R_{\text{m-iid}}(f) - R_{\text{emp}}(f)|$.

We analyze each component separately and then combine the results.

**Step 1. Bounding Standard *i.i.d.* Generalization Error.**

We define the supremum of the absolute difference between the true risk $R(f)$ and the empirical risk based on $m$ *i.i.d.* samples $R_{\text{m-iid}}(f)$:

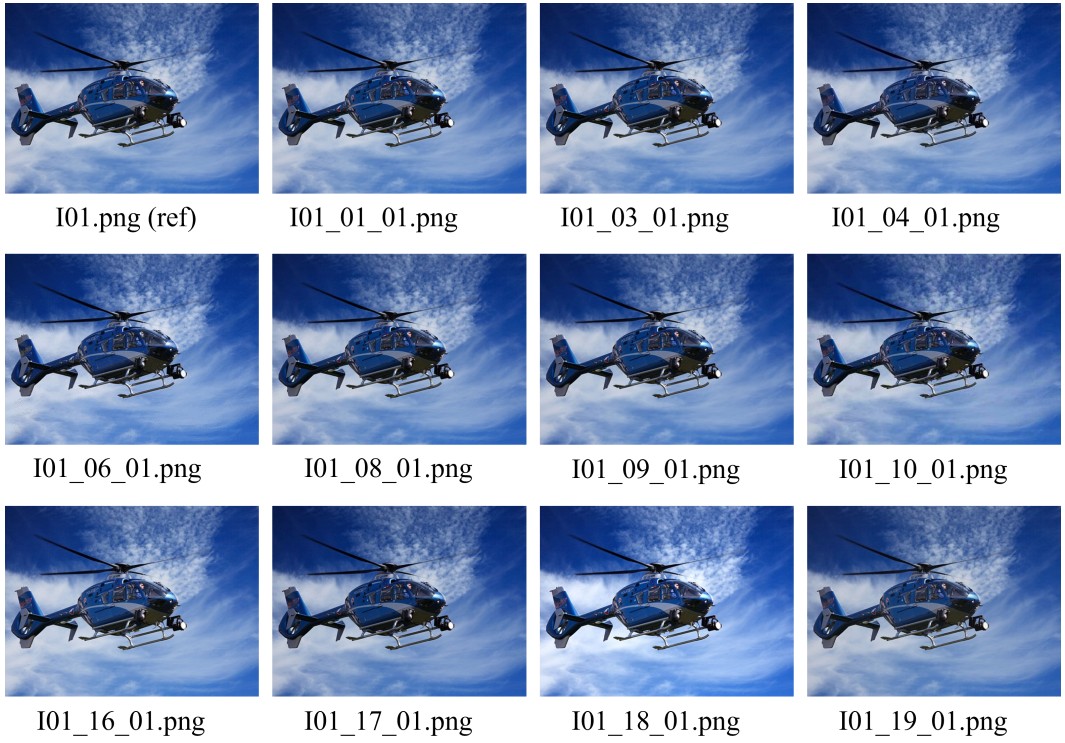

Figure 7: Part of low-distortion images in the KADID-10k dataset [38] using I01.png as reference. These images exhibit extremely high visual similarity.

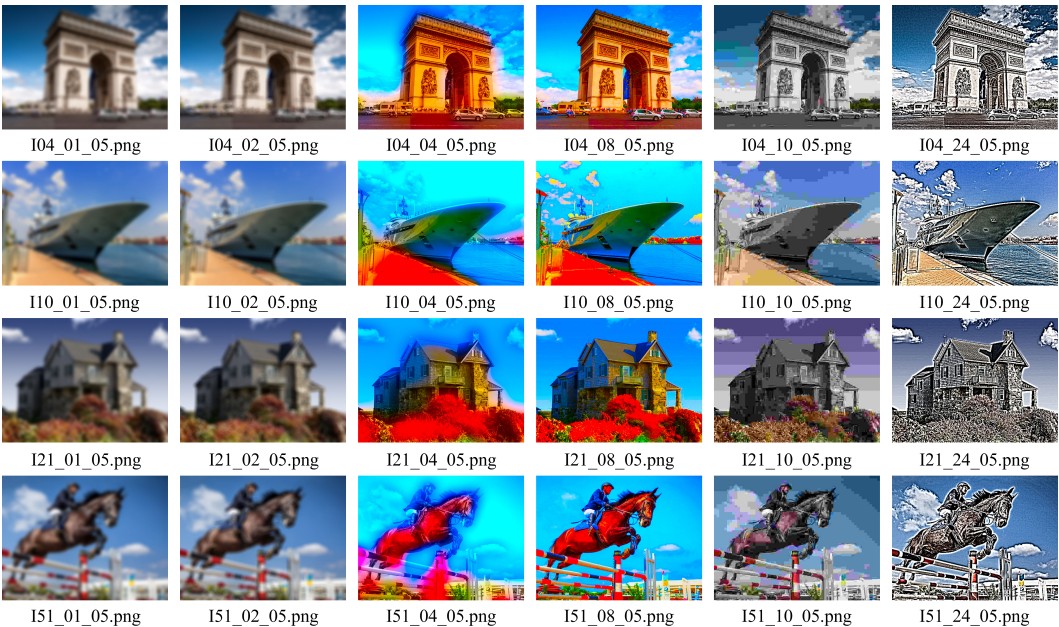

Figure 8: Part of high-distortion images in the KADID-10k dataset [38]. The same distortions exhibit very consistent patterns, and even different distortions may present very similar patterns.

$$\Phi(X) = \sup_{f \in \mathcal{F}} |R(f) - R_{\text{m-iid}}(f)|.$$

By applying McDiarmid's inequality, with probability at least $1 - \delta/2$:

$$\Phi(X) \leq \mathbb{E}[\Phi(X)] + \sqrt{\frac{2\log(2/\delta)}{m}},$$

Then, we upper bound the expectation using Rademacher complexity. Therefore, with probability at least $1 - \delta/2$:

$$\mathbb{E}[\Phi(X)] \leq 2 \, \text{Rad}_m(\mathcal{F}),$$

where $\text{Rad}_m(\mathcal{F})$ is the empirical Rademacher complexity based on $m$ *i.i.d* samples.

**Step 2: Bounding the Clustering Bias** We partition the dataset into $m$ cluster centers $\{x_1, x_2, \ldots, x_m\}$, each sampled independently from local distributions $\mathcal{D}_i \subset \mathcal{D}$. For simplicity, we assume that the $m$ *i.i.d.* cluster center samples are noise-free, i.e. for each $x_i$, the label is given by $\mathbb{E}[Y_i]$.

For each center $x_i$, we generate $k_i$ samples $\{(x_{i1}, y_{i1}), (x_{i2}, y_{i2}), \ldots, (x_{ik_i}, y_{ik_i})\}$ from its local distribution $\mathcal{D}_i$, we define a random variable $Y_i$ as the average loss over this cluster:

$$Y_i = \frac{1}{k_i} \sum_{j=1}^{k_i} \ell(f(x_i j), y_{ij}).$$

Since $\mathcal{Y} = [0, 1]$ and the loss function $\ell(\cdot)$ is L1, we have $l(f(x_i j), y_{ij}) \in [0, 1]$ and can upper bound the variance of $Y_i$:

$$\text{Var}(Y_i) \leq \frac{1}{4k_i}.$$

Due to the high correlation between samples within clusters, the inter-cluster error approximates $R_{\text{emp}}(f)$. At this point, applying Bernstein's inequality with the variance bound above, with probability at least $1 - \delta/2$,

$$|R_{\text{m-iid}}(f) - R_{\text{emp}}(f)| \approx \frac{1}{m} \sum_{i=1}^{m} (Y_i - \mathbb{E}[Y_i]) \leq \sqrt{\frac{\eta\log(2/\delta)}{8m}} + \frac{2\log(2/\delta)}{3m}, \qquad (3)$$

where $\eta = \frac{1}{m} \sum_{i=1}^{m} \frac{1}{k_i}$.

**Step 3. Combining Terms.**

Applying the union bound to the two components, with probability at least $1 - \delta$,

$$|R(f) - R_{\text{emp}}(f)| \leq 2 \, \text{Rad}_m(\mathcal{F}) + \sqrt{\frac{2\log(2/\delta)}{m}} + \sqrt{\frac{\eta\log(2/\delta)}{8m}} + \frac{2\log(2/\delta)}{3m}. \qquad (4)$$

$\square$

# D    Qualitative Analysis for Algorithm 1

To intuitively demonstrate the effect of Algorithm 1, we extracted features from the upsampled reference image set using an ImageNet-pretrained network (i.e., the feature extractor $f()$ used in Algorithm 1) and performed a UMAP visualization, as shown in Fig. 9.

From the overall distribution, it can be observed that the newly added reference samples are mainly distributed in the boundary zones or gap regions among the original training reference samples. This

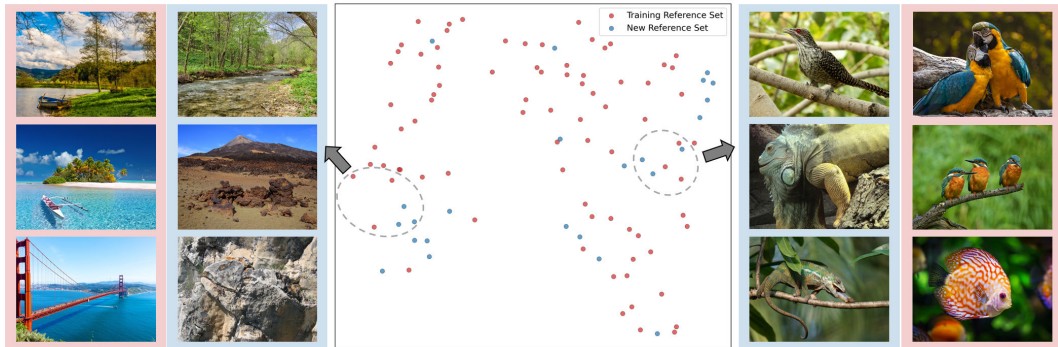

Figure 9: UMAP visualization of the reference image set after content upsampling by Algorithm 1, with features extracted via the ImageNet-pretrained network $f()$. Red dots denote the original training reference samples, while blue dots denote the newly selected samples produced by Algorithm 1. The left and right panels display image examples corresponding to two local regions.

indicates that Algorithm 1 effectively expands the coverage of the sample space without disrupting the original data distribution, thereby producing a more diverse and balanced reference set.

The two local regions on the left and right sides of Fig. 9 further illustrate that the newly added samples exhibit reasonable differences from the original ones in terms of content and visual characteristics. This demonstrates that Algorithm 1 successfully increases content diversity while maintaining distributional consistency and avoiding the introduction of redundant samples.

## E   Hyperparameter Ablation Analysis

To comprehensively evaluate the effect of key hyperparameters in our proposed framework, we perform a series of ablation experiments under the synthetic-to-authentic setting (KADID-10k $\rightarrow$ LIVEC, KonIQ-10k, and BID).

**Hyperparameter Analysis for DDCUp.** The DDCUp involves two key parameters, as defined in Algorithm 2: the number of nearest neighbors $k$, which determines the diversity of pseudo-label references, and $T_{rf}$, which controls the feature similarity threshold for selecting reliable neighbors. As shown in Table 5, when $k$ is too small, the pseudo-labels become unstable due to insufficient neighbor information, while when $k$ is too large, unreliable neighbors introduce noise and slightly degrade performance. Similarly, relaxing $T_{rf}$ from 0.05 to 0.1 causes a small drop in performance because less reliable neighbors are included. The best results are obtained when $k = 5$ and $T_{rf} = 0.05$, which strike a good balance between diversity and reliability.

Table 5: Ablation analysis of $k$ and $T_{rf}$ in DDCUp.

| SRCC | $k$ | $T_{rf}$ | LIVEC | KonIQ-10k | BID |
|------|-----|----------|-------|-----------|-----|
| Ours | 5 | 0.05 | **0.713** | **0.727** | 0.788 |
| 1 | 1 | – | 0.703 | 0.710 | 0.782 |
| 2 | 3 | 0.05 | 0.704 | 0.710 | 0.783 |
| 3 | 5 | 0.1 | 0.704 | 0.713 | 0.779 |
| 4 | 10 | 0.1 | 0.712 | 0.723 | 0.776 |
| 5 | ALL | – | 0.708 | 0.723 | **0.789** |

**Hyperparameter Analysis for DRCDown.** The DRCDown contains three key parameters, as described in Algorithm 3: the quality score threshold $T_g$, the feature difference threshold $T_{df}$, and the minimum retained sample number $T_u$. As shown in Table 6, $T_g$ and $T_{df}$ jointly control the redundancy detection process. If either $T_g$ or $T_{df}$ is too large, the model prunes similar samples too aggressively and may mistakenly remove diverse examples. Conversely, if $T_g$ or $T_{df}$ is too small, redundant samples are not detected effectively. The best results are obtained with $T_g = 1$

and $T_{df} = 0.1$, which yield the best performances by balancing effective redundancy reduction with sample diversity preservation.

Table 6: Ablation analysis of $T_g$ and $T_{df}$ in DRCDown.

| SRCC | $T_g$ | $T_{df}$ | LIVEC | KonIQ-10k | BID |
|------|-------|----------|-------|-----------|-----|
| Ours | 1 | 0.1 | **0.713** | **0.727** | 0.788 |
| 1 | 0.5 | 0.1 | 0.700 | 0.703 | 0.792 |
| 2 | 1.5 | 0.1 | 0.702 | 0.698 | **0.796** |
| 3 | 1 | 0.05 | 0.709 | 0.716 | 0.792 |
| 4 | 1 | 0.15 | 0.709 | 0.715 | 0.790 |
| 5 | 0.5 | 0.05 | 0.701 | 0.695 | 0.789 |
| 6 | 1.5 | 0.15 | 0.710 | 0.704 | 0.792 |

Table 7 reports the results under various $T_u$ values. A smaller $T_u$ results in excessive downsampling, losing critical information, while a larger $T_u$ retains substantial redundancy and diminishes the benefit of DRCDown. The model performs best with $T_u = 20$, where redundant clusters are sufficiently compact while maintaining adequate representation diversity.

Table 7: Ablation analysis of $T_u$ in DRCDown.

| SRCC | $T_u$ | LIVEC | KonIQ-10k | BID |
|------|-------|-------|-----------|-----|
| Ours | 20 | **0.713** | **0.727** | 0.788 |
| 1 | 10 | 0.703 | 0.709 | 0.777 |
| 2 | 30 | 0.701 | 0.694 | **0.791** |

From these results, we observe that the DRCDown module also demonstrates stable performance across a reasonable range of parameter values. Our final configurations ($T_g = 1$, $T_{df} = 0.1$, $T_u = 20$) consistently yield the best trade-off between redundancy suppression and dataset representativeness.

# F  Model Architecture and Pretraining Ablation

To thoroughly validate the generality and robustness of SynDR-IQA, we conduct ablation studies from two complementary perspectives: (1) the influence of different backbone architectures; and (2) the effect of distinct pretraining strategies, namely ImageNet and CLIP. These experiments collectively aim to assess whether SynDR-IQA's effectiveness depends on specific model architectures or feature initialization schemes.

## F.1  Impact of Backbone Architectures

To examine architecture generality, we evaluate SynDR-IQA with two representative transformer-based backbones: Vision Transformer (ViT-B/16) [39] and Swin Transformer Tiny (Swin-T) [38]. Both backbones follow the same training protocol as in our main experiments. The baseline corresponds to the DQGA.

As shown in Table 8, SynDR-IQA consistently improves the baseline performance across all authentic datasets, achieving an average SRCC gain of 7.2% with the Swin-T and 2.0% with ViT-B/16. These results indicate that SynDR-IQA effectively generalizes across diverse network architectures, confirming the robustness and versatility of our framework.

Table 8: Performance comparison between the baseline and SynDR-IQA using different ImageNet-pretrained backbones under the synthetic-to-authentic setting (SRCC only).

| Backbone | Method | LIVEC | KonIQ-10k | BID | Average |
|----------|--------|-------|-----------|-----|---------|
| Swin-T | Baseline | 0.620 | 0.600 | 0.729 | 0.649 |
| | SynDR-IQA | **0.670** | **0.719** | **0.777** | **0.721** |
| ViT-B/16 | Baseline | 0.714 | 0.694 | 0.740 | 0.716 |
| | SynDR-IQA | **0.729** | **0.717** | **0.761** | **0.736** |

### F.2 Impact of Pretraining Strategies

Beyond architectural variations, we further investigate the impact of different pretraining strategies on model performance. Specifically, we compare ImageNet pretraining with CLIP pretraining [40], which demonstrates strong semantic-level feature learning through large-scale vision-language pretraining.

We first explore whether replacing the backbone entirely with a CLIP-pretrained model can improve performance. Table 9 compares the CLIP-pretrained ViT-B/16 with the ImageNet-pretrained ViT-B/16 as the backbone, under identical architectural and training configurations.

Table 9: Comparison of ImageNet and CLIP pretraining when used as the backbone (SRCC only).

| Backbone | Method | LIVEC | KonIQ-10k | BID | Average |
|---|---|---|---|---|---|
| ViT-B/16 (ImageNet) | Baseline | 0.714 | 0.694 | 0.740 | 0.716 |
| | SynDR-IQA | **0.729** | 0.717 | 0.761 | **0.736** |
| ViT-B/16 (CLIP) | Baseline | 0.653 | 0.683 | 0.706 | 0.681 |
| | SynDR-IQA | 0.692 | **0.744** | **0.766** | 0.734 |

As shown in Table 9, replacing the backbone with CLIP does not yield a clear performance advantage over ImageNet pretraining. We attribute this to CLIP's multimodal pretraining objective, which emphasizes high-level semantic alignment between vision and language modalities at the expense of low-level visual details that are crucial for perceptual quality assessment tasks.

However, inspired by CLIP's superior semantic understanding capabilities, we further explore a more refined strategy: employing CLIP solely in the DDCUp module (as $f(\cdot)$), while retaining the ImageNet-pretrained ViT-B/16 as the primary IQA backbone. This design takes advantage of CLIP's superior semantic understanding in handling diverse content, while retaining the low-level visual representations in the main network necessary for accurate quality prediction.

Table 10: Performance comparison when using different encoders in the DDCUp module. The main IQA backbone remains ViT-B/16 (ImageNet-pretrained) (SRCC only).

| DDCUp Encoder | LIVEC | KonIQ-10k | BID | Average |
|---|---|---|---|---|
| ViT-B/16 (ImageNet) | 0.729 | 0.717 | 0.761 | 0.736 |
| ViT-B/16 (CLIP) | **0.739** | **0.751** | **0.778** | **0.756** |

As shown in Table 10, selectively integrating CLIP within the DDCUp module leads to substantial performance improvements, achieving new state-of-the-art results across all test datasets. We believe these gains stem from CLIP's enhanced semantic understanding, which enables more accurate content-aware reference selection during the upsampling process and further enhances the overall training effectiveness of the model.

## G Qualitative Results

To qualitatively demonstrate the effectiveness of our method, we showcase several representative examples from LIVEC in Fig. 10. The examples span diverse scenarios with various quality scores, distortions, scenes, and content. Notably, our model, trained solely on the synthetic distortion dataset KADID-10k, generates predictions that align well with human-annotated ground truth scores on these real-world images, validating the effective synthetic-to-real generalization capability of our approach. Furthermore, compared to the state-of-the-art method CLIP-IQA [33], our approach shows significantly better alignment with human perception.

## H Limitations

While SynDR-IQA demonstrates significant improvements in synthetic-to-algorithmic generalization scenarios, there are still notable performance gaps with practical availability. We think the primary challenge stems from the fundamental difference between existing synthetic distortion patterns and algorithmic distortion characteristics. Current synthetic distortion datasets primarily focus on

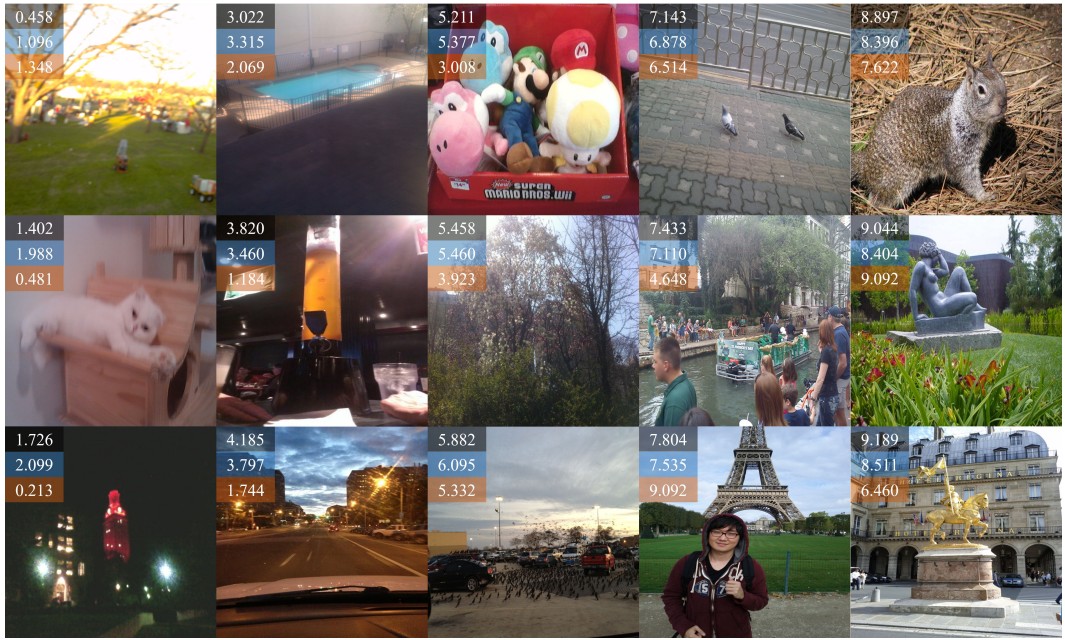

Figure 10: Qualitative results of SynDR-IQA on LIVEC. For each image, the top number represents the human-annotated ground-truth score, normalized to the range [0, 10]; the middle number represents our model's predicted quality score; and the bottom number represents the quality score predicted by CLIP-IQA [33]. The ground-truth scores of these images progressively increase from left to right and from top to bottom.

traditional degradation types (e.g., blur, noise, compression), which differ significantly from the complex patterns introduced by modern image processing algorithms, especially those involving deep learning-based methods.

This limitation highlights the need for synthetic distortion generation methods that can produce synthetic distortions with algorithmic distortion and other complex characteristics while maintaining controllable image quality degradation. We believe addressing this limitation through future research will be crucial for further improving the generalization capability of BIQA models across different distortion scenarios.

