# OpenReview forum: "Towards Syn-to-Real IQA: A Novel Perspective on Reshaping Synthetic Data Distributions"
_NeurIPS.cc/2025/Conference — NeurIPS 2025 poster_

### Official Review · Reviewer_5ayS · 2025-07-01

**Clarity:** 2
**Significance:** 2
**Originality:** 2
**Rating:** 4
**Confidence:** 5

**Summary:**

The authors propose a Blind Image Quality Assessment method employing a synthetic distortion to real distortion framework. The proposed method uses distribution-aware upsampling to obtain diverse reference contents, and density aware redundant cluster downsampling to reduce excessive redundant samples in the dataset. Experiments demonstrate that these strategies improve the performance (incrementally) across many IQA datasets.

**Questions:**

1. It is a bit confusing when authors interchangeably use synthetic dataset and synthetic distortions. Synthetic dataset can mean different things such as animation images, images generated through generative models etc., whereas in this context it primarily refers to natural images which are corrupted with synthetic distortions

2. Is there a way to evaluate Algorithm 1 alone? The goal was to enrich training data by increasing distribution diversity. Can we qualitatively or quantitatively evaluate if the proposed scheme is actually doing this?

3. Provide more intuitions on how Algorithm 1, 2 and 3 were designed. At a high level all these methods look like a bunch of 'if' conditions

4. The authors mention "only distortion intensities of levels 1, 3, and 5 are implemented". It would be interesting to know if all distortion levels were included and performance was evaluated. This will also throw light on how each distortion level influences the overall quality related feature learning.

**Ethical Concerns:**

["NO or VERY MINOR ethics concerns only"]

**Final Justification:**

The authors addressed my major concerns I had raised. The responses by the authors are convincing to improve my original score. I believe the novelty aspect of this paper is a bit incremental making it suitable for "Borderline Accept" recommendation.

**Limitations:**

The authors acknowledge the limitations in generalization capability between synthetic and authentic distortions. I believe there is one more limitation which is based on the assumption "similar content under the same distortion conditions should result in similar quality degradation".

**Quality:**

2

**Strengths And Weaknesses:**

Strengths
1. The only novelty I see in this work is that it proposes a refinement methodology to improve diversity and reduce redundancy in the training dataset. This strategy (although designed in a very ad-hoc manner) does provide some performance benefits especially for cross dataset generalization.

Weakness
1. Related Work is incomplete. The authors have missed out key references which employ distortion classification for BIQA task

a. W. Kim, A.-D. Nguyen, S. Lee, and A. C. Bovik, “Dynamic receptive field generation for full-reference image quality assessment,” IEEE Trans. Image Process., vol. 29, pp. 4219–4231, 2020.

b. Madhusudana, P. C., Birkbeck, N., Wang, Y., Adsumilli, B., and Bovik, A. C. Image quality assessment using
contrastive learning. IEEE Transactions on Image Processing, 31:4149–4161, 2022.

c. Saha, A., Mishra, S., and Bovik, A. C. Re-iqa: Unsupervised learning for image quality assessment in the wild. In
Proceedings of the IEEE/CVF conference on computer vision and pattern recognition, pp. 5846–5855, 2023

2. I find it hard to connect the proposed method with Theorem 3.1. The conclusion from the theorem is pretty generic and there is no BIQA specific concept. That theorem is applicable to any clustering scheme.

3. The Algorithm 1,2 and 3 look quite ad-hoc and not very intuitive. The authors use many heuristics in the algorithm without proper justification. For example in Algorithm 1 why minimum, maximum and median is used at different places? For algorithm 2 why distortion intensities of levels 1, 3, and 5 are used for all distortions? Shouldn't that be distortion specific?

4. The performance improvement is often incremental when compared to the current state-of-the-art. Also many IQA datasets have been skipped from reporting

a. Z. Ying, H. Niu, P. Gupta, D. Mahajan, D. Ghadiyaram, and A. Bovik, “From patches to pictures (PaQ-2-PiQ): Mapping the perceptual space of picture quality,” in Proc. IEEE Conf. Comput. Vision Pattern Recognit., 2020, pp. 3575–3585.

b. Y. Fang, H. Zhu, Y. Zeng, K. Ma, and Z. Wang, “Perceptual quality assessment of smartphone photography,” in Proc. IEEE Conf. Comput. Vision Pattern Recognit., 2020, pp. 3677–3686.

5. In the statement "Notably, to reduce the increase of redundant samples and prevent additional label noise, only distortion intensities of levels 1, 3, and 5 are implemented." What exactly are these levels? These levels can have different meaning for different distortions?

---

> ### Author Rebuttal · Authors · 2025-07-31
>
> Thank you very much for taking the time to review our work in such detail. We deeply appreciate each of your concerns and take your suggestions very seriously. We believe that the following detailed responses and additional experiments will effectively address your doubts and demonstrate the value and rigor of our work.
>
> ---
>
> **Weakness 1: Incomplete related work**
>
> Thank you for your suggestion. Reference [a] (Kim et al., 2020) primarily addresses full-reference IQA, which is significantly different from our blind IQA setting. References [b] (Madhusudana et al., 2022) and [c] (Saha et al., 2023) both focus on self-supervised IQA,  which differs somewhat from our cross-domain setting. Nevertheless, we will appropriately discuss and cite these works in the revised Related Work section to provide a more comprehensive and balanced overview.
>
> ---
>
> **Weakness 2:  Connection between Theorem 3.1 and proposed method**
>
> Thank you for your comments. We agree that Theorem 3.1 is indeed general and can be applied to other tasks with similar data distribution characteristics. Our contribution is that, for the first time, we explicitly identify discrete clustering as the fundamental cause of syn-to-real generalization failure in IQA, and analyze its impact through theoretical derivation. However, while the theorem itself is universal, the translation from theory to practical solutions is task-specific. For instance, to address insufficient diversity, we leveraged the IQA prior that "similar content under the same distortion conditions should result in similar quality degradation" to design the DDCup module. For the issue of redundancy imbalance, we specifically considered the common occurrence of hard samples with slight feature differences but significant quality differences in synthetic IQA datasets.
>
> In fact, your point that "the theorem is applicable to any clustering scheme" further highlights the value of our work: our theoretical framework and methodological insights can inspire solutions for other vision tasks that share similar data distribution properties.
>
> ---
>
> **Weakness 3 (Part) & Question3: Algorithm design intuitions**
>
> We appreciate the reviewer's valuable feedback. We would like to clarify that our proposed algorithms are not ad-hoc creations but are designed based on a deep understanding of the IQA task's intrinsic properties and are theoretically guided by Theorem 3.1 in our paper (increasing content diversity (m) and balancing sample density (by decreasing η) to enhance the generalization performance of the BIQA model).
>
> **Algorithm 1**:
> The core principle behind Algorithm 1 is to enrich the diversity of the training set while ensuring data reliability. The algorithm is carefully designed to address two primary issues:
> 1. **Redundant content:** Avoiding the selection of new samples that are overly similar either to existing samples or to one another, which would result in new redundant samples.
> 2. **Noisy outliers:** Preventing the introduction of samples that are excessively different from the labeled data, which could shift the data distribution and, more importantly, violate the assumption about similar content thereby compromising the reliability of the generated pseudo-labels.
>
> To this end, Algorithm 1 employs the following constraints:
> - `Min(DistN) > Median(DistT)` (Line 7) : Ensures that new reference images are sufficiently distinct from each other, thus reducing redundancy.
> - `Min(DistC) > Median(DistT)`(Line 5) : Ensures the new reference images are not too close to any existing reference image, avoiding overlap with the original dataset.
> - `Max(DistC) < Max(DistT)`(Line 5) : Restricts the maximum distance between any new and existing reference image, ensuring all new samples are still within an appropriate content distribution range, and thus reliable for pseudo-label generation.
>
> **Algorithm 2**:
> The design of this algorithm is based on a fundamental principle in the IQA field: the perceptual quality of an image is jointly determined by its content and distortion. From this, we derive a key hypothesis that similar content under the same distortion conditions should result in similar quality degradation. Based on this hypothesis, our strategy is both intuitive and logical: we generate pseudo-labels by finding content-similar neighboring images and referencing their quality scores under same distortion conditions.
>
> **Algorithm 3**:
> Algorithm 3 (DRCDown) aims to balance sample density (decreasing η) by downsampling redundant samples in synthetic IQA datasets. Specifically, it uses both feature distance and label distance (Line 3) to ensure that only samples highly similar in both visual content and quality score are removed. This approach effectively distinguishes truly redundant samples from hard samples with slight feature differences but significant quality differences (which is common in synthetic IQA datasets), thereby preserving valuable training information for the model.
>
>
> ---
>
> **Weakness 3 (Part) & Weakness 5 & Question 4: Distortion level selection**
>
> All synthetic IQA datasets provide multiple distortion strength settings for each distortion type. Specifically, in KADID-10k and KADIS-700k, each distortion type is available at 5 levels, each with specific parameter configurations. Our choice of levels 1, 3, and 5 is primarily to reduce data redundancy (see Page 4, Lines 124–130 for details) and to decrease the training cost.
>
> Following your suggestion, we further conducted experiments including all levels 1–5. The results are as follows:
>
> |                     | LIVEC (SRCC/PLCC) | KonIQ-10k (SRCC/PLCC) | BID (SRCC/PLCC) |
> | ------------------- | ----------------- | --------------------- | --------------- |
> | Levels 1,3,5 (Ours) | **0.713/0.705**   | **0.727/0.722**       | 0.788/0.769     |
> | Levels 1-5 (Full)   | 0.697/0.705       | 0.702/0.717           | **0.792/0.771** |
>
> The results show that including all levels leads to a slight improvement on BID, but causes a performance drop on LIVEC and KonIQ-10k. This confirms that introducing excessive samples with similar distortion strength increases redundancy and can harm the model's generalization ability.
>
> ---
>
> **Weakness 4: Performance improvements and dataset coverage**
>
> Thank you for your comments. Our method achieves notable improvements on three real-distortion datasets for cross-domain generalization from synthetic to authentic distortion, as reported in Table 1. Specifically, on average, our approach improves SRCC by 0.027 and PLCC by 0.027 compared to the second-best method.
>
> With respect to the additional datasets mentioned by the reviewer ([a] PaQ-2-PiQ and [b] SPAQ), we note that, to the best of our knowledge, existing methods rarely report cross-domain results (synthetic→authentic) on these datasets. For this reason, we did not originally include them in our evaluation. Nevertheless, we have now conducted additional experiments on SPAQ and compared our results against the SOTA method DGQA (reproduced):
>
> | Method            | SPAQ (SRCC/PLCC) |
> | ----------------- | ---------------- |
> | DGQA (reproduced) | 0.787/0.777      |
> | SynDR-IQA (Ours)  | **0.809/0.801**  |
> These results on SPAQ further confirm the consistent advantage of our method.
>
>
> ---
>
> **Q1: Terminology confusion**
>
> Thank you for your comment. In the IQA field, the term "synthetic dataset" commonly refers to IQA datasets generated using synthetic distortions, as seen in related literature [1], [2]. However, to avoid confusion, we will consistently use "synthetic IQA dataset" in the revised manuscript for clarity.
>
> [1] Shi J, Gao P, Qin J. Transformer-based no-reference image quality assessment via supervised contrastive learning[C]//Proceedings of the AAAI conference on artificial intelligence. 2024, 38(5): 4829-4837.
>
> [2] Yang J, Fu J, Zhang Z, et al. Align-IQA: aligning image quality assessment models with diverse human preferences via customizable guidance[C]//Proceedings of the 32nd ACM International Conference on Multimedia. 2024: 10008-10017.
>
> ---
>
> **Q2: Evaluating Algorithm 1**
>
> Thank you for this insightful suggestion. In the revised version, we will include UMAP visualizations and representative example images in the appendix to qualitatively show how Algorithm 1 enriches the training data and increases distributional diversity. Due to NeurIPS rebuttal policies, we are currently unable to provide these visualizations and examples at this stage, but we will add them in the final version.
>
>
> ---
>
> We sincerely hope that these detailed explanations and additional experiments can address your concerns. While there is still room for improvement in some aspects of our work, we firmly believe in its value and contribution to addressing the important issue of syn-to-real generalization in the IQA field. We kindly ask you to reconsider our work based on these clarifications, and we greatly appreciate your constructive feedback.

---

> ### Comment · Area_Chair_3xMY · 2025-08-08
>
> Hi Reviewer 5ayS,
>
> This is a gentle reminder to participate in the discussion with the authors regarding their rebuttal. Your input at this stage is important and appreciated.
>
> Best,
> AC

---

### Official Review · Reviewer_eiCt · 2025-07-02

**Clarity:** 2
**Significance:** 2
**Originality:** 3
**Rating:** 4
**Confidence:** 4

**Summary:**

The paper shows that that synthetic IQA datasets tend to form discrete, clustered feature spaces, high-quality images are clustered by reference, low-quality by distortion type, and medium-quality samples generally scatter inconsistently which undermines regression performance on authentic, real-world images.

To overcome this, the papers propose a new method called SynDR-IQA, constituting of DDCUp to selectively up-sample underrepresented content and DRCDown to prune over-dense clusters. Results show that the proposed method improves over compared methods.

**Questions:**

1. Effect of Encoders: More recent pre-trained vision encoders like CLIP/DINO/DINOv2/SigLIP could be tried.
2. Please comment on the algorithmic complexity/Runtime of the proposed algorithms.
3. The results in Table 3 could be made more convincing with addition of statistical significance results.

There are major concerns I have. Addressing these partialy or fully can help change my decision on the paper.

**Ethical Concerns:**

["NO or VERY MINOR ethics concerns only"]

**Final Justification:**

The authors have conducted experiments that answer some of my concerns and helped me understand the advantages of the proposed method better. Considering the other reviews and responses to them, I am willing to increase my score by one point.

**Limitations:**

Some limitations are discussed. There are no expected potential negative societal implications of this work.

**Quality:**

2

**Strengths And Weaknesses:**

Strengths:
1. The authors identify a very important problem in the BIQA space. The authors identify synthetic distortions often cluster unnaturally in feature space and argue that can hurt regression-based IQA models.
2. Relatively good number of experiments presented in this paper, along with some ablations.

Weaknesses:
1. Experiments use a supervised pre-trained ResNet-50 (and some expts with one Swin transformer variant). The paper does not justify why using a ImageNet pretrained network is a reliable estimate of content representation of "real world" images? It is strange that in 2025 the authors did not consider using more recent pre-trained vision encoders like CLIP/DINO/DINOv2/SigLIP or even self supervised imagenet pre-trained encoders like MoCO/SimCLR/JEPA/BYOL.
2. There are inherent biases associated with using a pre-trained encoders.
3. Algorithm complexity/Runtime : There is no discussion on algorithmic complexities or runtime requirements for Algo 1-3. Also there are no intuitions on how the algorithms are designed, especially algorithm 1. For example, the authors claim in the paper  "original content distribution while also being sufficiently distinct from each other (lines 5 and 7)," there seems to be missing inspiration why and how this design was reached.
4. Some of the results reported in Table 3 show really small gains (cross database). It would be great if some sort of statistical significance test is performed to confirm of definite improvement. One option would be repeat the training for both the baseline and proposed 10 times or similar.
5. Also the authors should also compare their method with the popular family of un/self-supervised methods developed specially for IQA like CONTRIQUE, Re-IQA, ARNIQA etc.

---

> ### Author Rebuttal · Authors · 2025-07-31
>
> We sincerely appreciate your insightful and challenging comments. Your professional review precisely targeted the key aspects of our work and provided us with invaluable directions for improvement. We fully understand your concerns and have conducted extensive additional experiments and clarifications as per your suggestions. We believe these new results thoroughly address your questions and strongly demonstrate the value of our work.
>
> ---
>
> **Weakness 1 & Weakness 2 & Question 1: Choice of pre-trained vision encoders**
>
> We sincerely appreciate the reviewer’s valuable comments. In response to your suggestion, we have conducted additional experiments to compare the performance of CLIP as a pre-trained vision encoder (vit_base_patch16_clip_224.openai) with that of an ImageNet pre-trained encoder with the same architecture (vit_base_patch16).
>
> | Backbone            | Method    | LIVEC (SRCC/PLCC) | KonIQ-10k (SRCC/PLCC) | BID (SRCC/PLCC) |
> | :------------------ | :-------- | :---------------- | :-------------------- | :-------------- |
> | ViT-B/16 (ImageNet) | Baseline  | 0.714 / 0.737     | 0.694 / 0.732         | 0.740 / 0.740   |
> |                     | SynDR-IQA | 0.729 / 0.749     | 0.717 / 0.756         | 0.761 / 0.748   |
> | ViT-B/16 (CLIP)     | Baseline  | 0.653 / 0.662     | 0.683 / 0.738         | 0.706 / 0.709   |
> |                     | SynDR-IQA | 0.692 / 0.714     | 0.744 / 0.748         | 0.766 / 0.771   |
>
> The results indicate that using CLIP as the main vision encoder does not bring a clear performance advantage. We suspect this is because CLIP’s multimodal pretraining emphasizes semantic-level alignment, which may come at the cost of low-level visual details crucial for IQA tasks.
>
> However, your suggestion inspired us to further explore new uses for CLIP. Leveraging its strong semantic understanding, we employed CLIP encoder only in the DDCUp module, which only focuses on content, while the IQA backbone remained ViT-B/16 pretrained on ImageNet.
>
> | Method        | DDCUp Backbone       | LIVEC (SRCC/PLCC) | KonIQ-10k (SRCC/PLCC) | BID (SRCC/PLCC)   |
> | :------------ | :------------------- | :---------------- | :-------------------- | :---------------- |
> | SynDR-IQA<br> | ViT-B/16  (ImageNet) | 0.729 / 0.749     | 0.717 / 0.756         | 0.761 / 0.748     |
> |               | ViT-B/16 (CLIP)      | **0.739 / 0.751** | **0.751 / 0.784**     | **0.778 / 0.769** |
>
> It is clear that applying CLIP as the encoder in the DDCUp module significantly enhances performance, achieving new SOTA results. We believe this improvement is mainly due to the superior semantic understanding of CLIP, which enables more accurate selection of content-relevant reference images, thereby improving the quality of references used in upsampling and generating more reliable pseudo-labels for training.
>
> Thank you again for your insightful suggestions!
>
> ---
>
> **Weakness 3 & Question 2: Algorithm complexity/runtime and design inspiration**
>
> Thank you for your valuable comments. Regarding algorithmic complexity and runtime, our method does introduce some additional computational overhead; however, this overhead is entirely confined to the training phase, and there is no extra cost during inference.
>
> Specifically, excluding the generation of distorted images (which are pre-generated before training), the entire content upsampling process (DDCUp) takes only about 14.5 seconds. Moreover, for a given backbone and training set, content upsampling is a one-time preprocessing step. By adding the new training samples (pre-generated by DDCUp) into the training set and introducing DCRDown, the average per-epoch training time increases from 35.6 seconds to 55.2 seconds. Given that this additional computation results in significant improvements in model generalization, we believe this is a well-justified trade-off.
>
> Regarding the motivation behind Algorithm 1, our core principle is to **enrich the diversity of the training set while ensuring data reliability**. Thus, the algorithm is carefully designed to address two primary issues:
> 1. **Redundant content:** Avoiding the selection of new samples that are overly similar either to existing samples or to one another, which would result in new redundant samples.
> 2. **Noisy outliers:** Preventing the introduction of samples that are excessively different from the labeled data, which could shift the data distribution and, more importantly, violate the assumption about similar content thereby compromising the reliability of the generated pseudo-labels.
>
> To this end, Algorithm 1 employs the following constraints:
> - `Min(DistN) > Median(DistT)` (Line 7) : Ensures that new reference images are sufficiently distinct from each other, thus reducing redundancy.
> - `Min(DistC) > Median(DistT)`(Line 5) : Ensures the new reference images are not too close to any existing reference image, avoiding overlap with the original dataset.
> - `Max(DistC) < Max(DistT)`(Line 5) : Restricts the maximum distance between any new and existing reference image, ensuring all new samples are still within an appropriate content distribution range, and thus reliable for pseudo-label generation.
>
> We will provide more detailed explanations for these designs in the revised version of our manuscript.
>
> ---
>
> **Weakness 4 & Quesstion 3: Performance gain and statistical significance results**
>
> Thank you for this valuable suggestion. We understand the reviewer’s concern about verifying the reliability of performance improvements through statistical tests. First, we would like to clarify that the synthetic-to-synthetic experiments presented in Table 3 are not the core contribution of our work. Existing methods already demonstrate strong generalization capabilities on the synthetic-to-synthetic setting, and our improvements in this setting primarily serve to highlight the universality of our approach. The main contribution of our work lies in the more challenging synthetic-to-authentic transfer setup (Table 1), where the performance gains of our method are significant.
>
> Furthermore, in the field of IQA, statistical variance analysis is generally not applicable for cross-dataset evaluations. This is because the standard evaluation protocol for IQA uses fixed and complete datasets for training, without random data splits. This practice is also followed in recent works such as [1] and [2].
>
> [1] Li X, Gao T, et. al. Adaptive Feature Selection for No-Reference Image Quality Assessment by Mitigating Semantic Noise Sensitivity. Proceedings of the 41st International Conference on Machine Learning, in Proceedings of Machine Learning Research 235:27808-27821.
>
> [2] Zhong Y, Wu X, et. al. Causal-IQA: Towards the Generalization of Image Quality Assessment Based on Causal Inference. Proceedings of the 41st International Conference on Machine Learning, in  Proceedings of Machine Learning Research 235:61747-61762.
>
>
> ---
>
> **Weakness 5: Comparison with Self-supervised IQA Methods**
>
> Thank you for this valuable suggestion. Following your advice, we added comprehensive comparisons with leading self-supervised IQA methods such as CONTRIQUE and ARNIQA when trained on KADID-10k. The SRCC results are presented in the tables below. Due to the incomplete official implementation of Re-IQA, we were unable to reproduce its results.
>
> | Method           | LIVEC     | KonIQ-10k | BID       |
> | :--------------- | :-------- | :-------- | :-------- |
> | CONTRIQUE        | 0.350     | 0.419     | 0.322     |
> | ARNIQA           | 0.548     | 0.483     | 0.608     |
> | SynDR-IQA (Ours) | **0.713** | **0.727** | **0.788** |
> As shown above, our method consistently outperforms existing SOTA self-supervised IQA methods on various public benchmarks. Additionally, we find that our SynDR-IQA trained only on synthetic data achieves performance that is comparable to or even exceeds these methods trained on authentic datasets. This strongly demonstrates the effectiveness and robustness of our framework.
>
> | Method                  | Train     | LIVEC     | KonIQ-10k |
> | ----------------------- | --------- | --------- | --------- |
> | CONTRIQUE (Reprod.)     | LIVEC     | -         | 0.657     |
> |                         | KonIQ-10k | **0.744** | -         |
> | CONTRIQUE (Orig. Paper) | LIVEC     | -         | 0.676     |
> |                         | KonIQ-10k | 0.731     | -         |
> | ARNIQA                  | LIVEC     | -         | 0.684     |
> |                         | KonIQ-10k | 0.724     | -         |
> | SynDR-IQA (Ours)        | LIVEC     | 0.713     | -         |
> |                         | KonIQ-10k | -         | **0.727** |
>
> ---
>
> We sincerely hope that the detailed responses and additional experiments above will resolve all your concerns. We have made every effort to strengthen our manuscript, and we respectfully ask you to reconsider our work in light of this new evidence. Once again, thank you very much for your valuable time and expert guidance!

---

> > ### Comment · Reviewer_eiCt · 2025-08-02
> >
> > The authors have conducted additional experiments that directly address some of my concerns, which has clarified the strengths and advantages of the proposed method. In light of this and after considering the other reviewers' comments and the authors’ responses, I am inclined to increase my score by one point.
> >
> > I would encourage the authors to include all the results presented during the rebuttal in the final version paper, should the paper be accepted.

---

### Official Review · Reviewer_1mY6 · 2025-07-02

**Clarity:** 3
**Significance:** 3
**Originality:** 3
**Rating:** 3
**Confidence:** 4

**Summary:**

This paper addresses the generalization gap in Blind Image Quality Assessment (BIQA) between synthetic and real-world distortions. The authors propose SynDR-IQA, a framework to reshape synthetic data distributions by increasing content diversity and reducing sample redundancy. Two strategies are introduced: (1) Distribution-aware Diverse Content Upsampling (DDCUp) and (2) Density-aware Redundant Cluster Downsampling (DRCDown). A theoretical generalization bound is derived to justify the design choices. The method is evaluated on standard cross-dataset BIQA benchmarks and demonstrates improvements over prior work.

**Questions:**

• How sensitive is DDCUp to the threshold used for selecting pseudo-label neighbors? Would performance degrade if content distribution shifts?

• Why was DRCDown not compared with other sampling strategies (e.g., k-means undersampling, core-set selection, curriculum filtering)?

• Can you provide a variance analysis on key benchmark improvements?

**Ethical Concerns:**

["NO or VERY MINOR ethics concerns only"]

**Final Justification:**

While most of my concerns have been addressed, I feel the novelty is limited. I will as a result stick to my original rating.

**Limitations:**

yes

**Quality:**

3

**Strengths And Weaknesses:**

Strengths

- Targets a very important problem of limited generalization of synthetic datasets prevalent in BIQA

- The paper provides a theoretically grounded explanation for the observed failure mode, using a generalization bound for clustered data.

- The proposed techniques show measurable improvements over recent baselines on several cross-domain benchmarks.

- The ablation and visualization studies support the hypothesis.

Weakness

- Both proposed techniques (DDCUp and DRCDown) are relatively simple data augmentation and filtering heuristics. There is little innovation in model design or learning methodology.

- The pseudo-labeling strategy in DDCUp assumes that visually similar content leads to similar distortion behavior, which may not generalize beyond synthetic datasets. No evaluation is done to test this assumption under failure conditions.

- What is the reason for suboptimal performance of the approach in LIVEC and KONIQ-10k datasets as shown in Table 1 ?

- The paper lacks comparison of data downsampling approach with existing approaches such as clustering or data pruning baselines

---

> ### Author Rebuttal · Authors · 2025-07-31
>
> Thank you very much for your thorough review of our work and for the valuable comments you provided. Your professional feedback is extremely important for the improvement and refinement of our study. We understand your concerns and have provided detailed responses and clarifications to each point below.
>
> ---
>
> **Weakness 1:  Lack of model and learning innovation**
>
> We appreciate the reviewer's valuable comments. We agree that the implementation of DDCUp and DRCDown is fairly simple. However, we consider this simplicity a key strength of our work.
> Our main contribution is not a more complex model architecture. Instead, we are the first to provide a theoretically grounded and practical solution to the longstanding syn-to-real generalization challenge in BIQA, approached from the perspective of data distribution.
>
> Our key contributions can be summarized as follows:
>
> 1. Through visualization and theoretical analysis, we first reveal the fundamental causes behind the lack of generalization in existing IQA models: insufficient content diversity and sample redundancy in the training data jointly lead to clustering in the feature space. This insight creates a clear path for future improvements.
> 2. Based on this insight ,  we propose a framework to reshape data distribution with DDCUp and DRCDown, substantially enhancing model generalization. Additionally, as a data-based approach, our method requires no changes to existing model architectures and offers greater potential for practical deployment.
>
> In summary, we believe that the true innovation of our work lies in shifting the research paradigm from a focus on model architecture to a broader consideration of data construction and optimization. We believe such a simple yet effective approach carries significant practical value and inspiration for BIQA and related research fields.
>
> ---
>
> **Weakness2：The pseudo-labeling may not generalize beyond synthetic datasets**
>
>
> Thank you for your valuable feedback. We would like to first clarify the core assumption underlying our approach. You have characterized it as "visually similar content leads to similar distortion behavior," which may indeed cause some ambiguity.
>
> As described in our paper, our actual assumption is: "similar content under the same distortion conditions should result in similar quality degradation." We believe this assumption is reasonable as it builds upon a fundamental understanding in the IQA field: the perceptual quality of an image is jointly determined by its content and distortion. When distortion conditions are fixed, content becomes the primary factor influencing perceptual quality variations. Therefore, finding content-similar neighboring images and using their quality scores under the same distortion conditions as references to generate pseudo-labels represents a logically sound and well-grounded strategy, which is also supported by our experimental results.
>
> Furthermore, DDCUp is not designed as a universal pseudo-labeling scheme that generalizes to all unknown distortion types. Instead, it explicitly focuses on addressing the insufficient content diversity problem in existing synthetic distortion datasets, thereby improving model generalization in real-world distortion scenarios. Our entire experimental framework operates within the synthetic-to-real setting and makes no claims about extending beyond the scope of synthetic datasets.
>
> ---
>
> **Weakness 3: Performance on the LIVEC and KonIQ-10k Datasets**
>
> We appreciate the reviewer's detailed analysis of our results. We would like to clarify that our method is not "suboptimal" but rather demonstrates comprehensive and leading performance. In the overall evaluation across three test datasets, compared to the second-best method, our method achieves an average improvement of 0.027 in SRCC and 0.027 in PLCC, which significantly demonstrates the effectiveness of our approach.
>
> While we do not achieve the highest value on individual metrics for every dataset, this does not diminish our overall advantage. For example, in the KADID-10k→KonIQ-10k task, our SRCC is slightly lower than FreqAlign, but our PLCC metric surpasses it. More importantly, compared to FreqAlign, we achieves significant improvements across all three datasets with SRCC and PLCC increased by 0.063 and 0.064,  respectively.
>
> ---
>
> **Weakness 4 & Question 2: Comparison with Other Downsampling Approaches**
>
> Thank you for this valuable suggestion. To demonstrate the effectiveness of our DRCDown method, we have conducted additional experiments comparing it with Core-set Selection (K-center Greedy-based) method. The SRCC results are shown in the following table:
>
> | Method                       | LIVEC     | KonIQ-10k | BID       |
> | :--------------------------- | :-------- | :-------- | :-------- |
> | Baseline                     | 0.696     | 0.681     | 0.770     |
> | Core-set (ratio=0.8)         | 0.683     | 0.675     | 0.738     |
> | Core-set (ratio=0.6)         | 0.682     | 0.666     | 0.760     |
> | DRCDown                      | 0.690     | 0.711     | 0.768     |
> | DDCUp + Core-set (ratio=0.8) | 0.689     | 0.709     | 0.780     |
> | DDCUp + Core-set (ratio=0.6) | 0.697     | 0.705     | 0.769     |
> | DDCUp + DRCDown (Ours)   | **0.713** | **0.727** | **0.788** |
>
> The results clearly show that our DRCDown method consistently achieves better performance. The main reason is that Core-set primarily focuses on maintaining sample diversity and may accidentally remove important samples that are visually similar yet have large label differences (common in synthetic IQA datasets). In contrast, DRCDown accurately identifies and removes only  truly redundant samples that are visually similar and have similar quality scores. This helps retain more informative samples for model training and avoids the loss of valuable information.
>
> ---
>
> **Question 1: How sensitive the threshold used for selecting pseudo-label neighbors**
>
> We appreciate the reviewer’s thoughtful questions regarding our method’s sensitivity to the pseudo-label threshold.
>
> To address the first point, for Algorithm 2,  the key hyperparameters are $k$ (the number of nearest neighbors for pseudo-label generation) and $T_{rf}$​ (the maximum allowed feature similarity difference for reliable neighbors). We conducted additional ablation studies, varying $k$ and $T_{rf}$ across a broad range, and summarized the SRCC results in the table below:
>
> | SRCC | $k$ | $T_{rf}$ | LIVEC     | KonIQ-10k | BID       |
> | ---- | --- | -------- | --------- | --------- | --------- |
> | Ours | 5   | 0.05     | **0.713** | **0.727** | 0.788     |
> | 1    | 1   | \        | 0.703     | 0.710     | 0.782     |
> | 2    | 3   | 0.05     | 0.704     | 0.710     | 0.783     |
> | 3    | 5   | 0.1      | 0.704     | 0.713     | 0.779     |
> | 4    | 10  | 0.1      | 0.712     | 0.723     | 0.776     |
> | 5    | ALL | \        | 0.708     | 0.723     | **0.789** |
>
> As shown, our method maintains stable and competitive performance under a variety of hyperparameter choices, indicating robustness rather than high sensitivity to these settings. This demonstrates that both $k$ and $T_{rf}$ can be selected within reasonable ranges without significant degradation in results.
>
> ---
>
> **Question 3: Provide a variance analysis**
>
> Thank you very much for this question. In the field of IQA, variance analysis is generally not feasible for cross-dataset evaluation settings. This is because the standard evaluation protocol for IQA uses fixed and complete datasets for training, without random data splits. This practice is also followed in recent works such as [1] and [2].
>
> [1] Li X, Gao T, et. al. Adaptive Feature Selection for No-Reference Image Quality Assessment by Mitigating Semantic Noise Sensitivity. Proceedings of the 41st International Conference on Machine Learning, in Proceedings of Machine Learning Research 235:27808-27821.
>
> [2] Zhong Y, Wu X, et. al. Causal-IQA: Towards the Generalization of Image Quality Assessment Based on Causal Inference. Proceedings of the 41st International Conference on Machine Learning, in Proceedings of Machine Learning Research 235:61747-61762.
>
> ---
>
> We sincerely hope that the above clarifications and the additional experimental results address your concerns. We believe that our work offers a novel, effective, and solid perspective for tackling the generalization challenge in BIQA, and we kindly ask you to reconsider the value of our contributions.

---

> > ### Comment · Reviewer_1mY6 · 2025-08-06
> >
> > I thank the authors for their detailed response.
> > While the authors have addressed most of my concerns I am still unconvinced about few things.
> > 1. My concern about innovation remains partially unresolved. while i agree identification of root causes through visualization and theory is valuable, the proposed solutions still feel like straight forward applications of existing data augmentation principles.
> > 2. I accept your explanation about overall performance improvement. However, inconsistent performance across datasets suggests that the method may not be universally robust. This may mean that there are dataset specific characteristics that your approach does not fully capture.

---

> > > ### Author Response · Authors · 2025-08-07
> > >
> > > Thank you very much for your careful review and insightful feedback on our paper. We greatly appreciate the time and effort you have invested, and we value the opportunity to further clarify and elaborate on the key aspects of our work.
> > >
> > > ---
> > >
> > > We fully understand your concerns regarding the innovation of our method. We agree that the fundamental ideas behind our proposed strategies share certain similarities with existing data augmentation methods. However, we would like to emphasize that **the main innovations of this work go beyond the specific strategies themselves. They lie in the identification of core problems in the field, a systematic theoretical explanation, and the development of targeted methods based on these insights**.
> > >
> > > **Firstly, our key contribution lies in fundamentally revealing the core challenges of syn-to-real generalization in BIQA.** Through visualization and theoretical analysis, we are the first to clearly show that the insufficient content diversity in synthetic IQA datasets, together with local sample redundancy arising from data synthesis mechanisms, leads to discrete clusters in the feature space and ultimately hinder generalization. We believe that **clearly identifying and theoretically analyzing the essence of this long-standing challenge offers value comparable to introducing entirely new methods**. This theoretical insight establishes a solid foundation for future research and method optimization, and also brings a novel perspective to the field by encouraging a shift in focus from model optimization to data optimization. Additionally, we agree that future integration with techniques such as data pruning or data augmentation may further improve model performance.
> > >
> > > **Secondly, our methods are specifically designed based on these theoretical findings and tailored to the characteristics of IQA tasks, rather than simply applying general data augmentation techniques.** DDCUp is designed to enhance content diversity, with its synthesis approach tailored to the specific characteristics of the IQA task, fundamentally distinguishing it from general data augmentation methods. DRCDown aims to reduce local sample redundancy by focusing on retaining the hard samples (i.e., those with similar features but disparate labels) common in IQA datasets. The performance comparison with the classical data pruning method further validates the effectiveness of our design (see Rebuttal to Weakness 4 & Question 2).
> > >
> > > In summary, our work offers a new theoretical perspective and a specialized solution to the generalization challenge in IQA, rather than merely proposing a new technique. We believe that such advancements in theory and research perspective will have a positive and persistent impact on the field.
> > >
> > > ---
> > >
> > > We also appreciate your concerns regarding the robustness of our method. We would like to clarify that the observed inconsistency in performance across different datasets is largely due to the inherent differences in the baseline capabilities of various backbones on these datasets. **When we adopt ViT-B/16 as the backbone, our method achieves the best results across all datasets and evaluation metrics** (see Rebuttal to Weakness 1, Weakness 2, and Question 1 of Reviewer eiCt).
> > >
> > > It is important to emphasize that **our approach focuses on improving the distribution of the training dataset and is essentially independent of specific model architectures or target datasets.** Although there are inherent differences among backbones and datasets, our method consistently brings significant performance gains in all tested settings. These results demonstrate that our method effectively optimizes data distribution, enabling the model to learn more essential and generalizable quality representations, thus leading to consistent improvements across a wide range of scenarios.
> > >
> > > ---
> > >
> > > We again thank you for your constructive comments. We hope that our detailed responses above address your concerns, clarify the novelty and impact of our contributions, and demonstrate the robustness and potential of our method. We look forward to any further feedback and are committed to advancing research in this important area.

---

### Official Review · Reviewer_57Bm · 2025-07-02

**Clarity:** 4
**Significance:** 3
**Originality:** 4
**Rating:** 5
**Confidence:** 5

**Summary:**

This paper tackles the challenge of synthetic-to-real generalization in BIQA. The authors make a key observation that models trained on synthetic data tend to produce discrete, clustered feature representations that hinder generalization. Through theoretical analysis, they attribute this limitation to insufficient content diversity and sample redundancy. To overcome these challenges, they propose SynDR-IQA, a data-centric framework with two novel strategies (DDCup and DRCDown) that reshapes the training data distribution to improve generalization. Extensive experiments across multiple cross-dataset settings validate the framework's effectiveness.

**Questions:**

Please refer to the weakness.

**Ethical Concerns:**

["NO or VERY MINOR ethics concerns only"]

**Limitations:**

Yes

**Quality:**

4

**Strengths And Weaknesses:**

Strength
1. The paper provides a novel and insightful perspective on why synthetic-to-real generalization fails in IQA, providing rigorous theoretical analysis of how sample diversity and redundancy affect generalization bounds.
2. The proposed DDCup and DRCDown strategies directly address the identified clustering problems through a well-motivated, data-centric approach that requires no architectural changes.
3. Consistent improvements across multiple cross-dataset evaluation settings demonstrate the effectiveness and broad applicability of the framework.
4. The paper is clearly written and well-organized.

Weakness
1. The proposed DDCup and DRCDown strategies rely on several hyperparameters whose sensitivity is not thoroughly analyzed.
2. More visualization results on other synthetic and real-world datasets should be provided to better support the main claims.
3. The UMAP visualization for the model "Trained directly on KADID-10k" should also be added to Figure 3 to better illustrate the limitation of the original synthetic dataset.

---

> ### Author Rebuttal · Authors · 2025-07-31
>
> We sincerely thank you for your thorough review, encouraging assessment, and insightful suggestions. We are greatly inspired by your constructive feedback, which is invaluable for enhancing the quality of our paper. We have conducted targeted supplementary experiments and revised the manuscript accordingly. Our point-by-point responses are detailed below:
>
> ---
>
> **Weakness 1: Hyperparameter sensitivity analysis**
>
> We appreciate the insightful comments regarding the sensitivity of our method to hyperparameters. In response, we conducted additional ablation studies and have incorporated these analyses into the revised manuscript.
>
> For Algorithm 2, the key hyperparameters are $k$ (the number of nearest neighbors for pseudo-label generation) and $T_{rf}$​ (the maximum allowed feature similarity difference for reliable neighbors). We tested various combinations of these parameters and summarized the SRCC results as follows:
>
> | SRCC | $k$ | $T_{rf}$ | LIVEC     | KonIQ-10k | BID       |
> | ---- | --- | -------- | --------- | --------- | --------- |
> | Ours | 5   | 0.05     | **0.713** | **0.727** | 0.788     |
> | 1    | 1   | \        | 0.703     | 0.710     | 0.782     |
> | 2    | 3   | 0.05     | 0.704     | 0.710     | 0.783     |
> | 3    | 5   | 0.1      | 0.704     | 0.713     | 0.779     |
> | 4    | 10  | 0.1      | 0.712     | 0.723     | 0.776     |
> | 5    | ALL | \        | 0.708     | 0.723     | **0.789** |
> For Algorithm 3, we analyzed three key hyperparameters: $T_g$​ (quality score threshold), $T_{df}​$ (feature difference threshold), and $T_u$​ (minimum retained sample number after downsampling). The corresponding results are:
>
> | SRCC | $T_g$ | $T_{df}$ | LIVEC     | KonIQ-10k | BID       |
> | ---- | ----- | -------- | --------- | --------- | --------- |
> | Ours | 1     | 0.1      | **0.713** | **0.727** | 0.788     |
> | 1    | 0.5   | 0.1      | 0.700     | 0.703     | 0.792     |
> | 2    | 1.5   | 0.1      | 0.702     | 0.698     | **0.796** |
> | 3    | 1     | 0.05     | 0.709     | 0.716     | 0.792     |
> | 4    | 1     | 0.15     | 0.709     | 0.715     | 0.790     |
> | 5    | 0.5   | 0.05     | 0.701     | 0.695     | 0.789     |
> | 6    | 1.5   | 0.15     | 0.710     | 0.704     | 0.792     |
>
> | SRCC | $T_u$ | LIVEC     | KonIQ-10k | BID       |
> | ---- | ----- | --------- | --------- | --------- |
> | Ours | 20    | **0.713** | **0.727** | 0.788     |
> | 1    | 10    | 0.703     | 0.709     | 0.777     |
> | 2    | 30    | 0.701     | 0.694     | **0.791** |
>
> ---
> **Weakness 2: More visualization results on additional datasets**
>
> Thank you for this valuable suggestion. To comprehensively support our main arguments, we have generated feature distribution visualizations on additional synthetic (TID2013) and real-world (BID, KonIQ-10k) datasets. These new visual results are consistent with our conclusions. Due to NIPS rebuttal policies, we cannot submit new figures directly in the response. However, we assure you that these visualizations will be incorporated into the final version of the paper.
>
> ---
>
> **Weakness 3: UMAP visualization of baseline model in Fig. 3**
>
> Thank you for this excellent suggestion. As you proposed, adding a visualization for the baseline model (i.e., trained "directly on KADID-10k") to Fig. 3 highlights the advantages of our method. The baseline's features form distinct, scattered clusters (with an SRCC of only 0.5449), which stands in stark contrast to the continuous and smooth feature distribution achieved by our method. This addition makes the generalization advantage of our approach immediately clear. The updated figure will be included in the final paper.
>
> ---
>
> Thank you again for your valuable time and insightful feedback. Your comments have been instrumental in improving our work, and have significantly enhanced the rigor and clarity of our paper. We sincerely hope that our responses and revisions have fully addressed your concerns.

---

> ### Comment · Area_Chair_3xMY · 2025-08-08
>
> Hi Reviewer 57Bm,
>
> This is a gentle reminder to participate in the discussion with the authors regarding their rebuttal. Your input at this stage is important and appreciated.
>
> Best,
> AC

---

### Official Review · Reviewer_BpnP · 2025-07-03

**Clarity:** 2
**Significance:** 2
**Originality:** 3
**Rating:** 5
**Confidence:** 4

**Summary:**

This paper is about improving the generalizability of no-reference image quality assessment methods that are trained on limited data. The authors provide a key observation that representations learned from synthetic data tend to present a discrete and clustered pattern, and this issue is due to the distribution of synthetic data, especially the given distortion types. To solve this problem, the paper proposes a novel framework called SynDR-IQA to reshape the synthetic data distribution. Specifically, after theoretical analysis, the framework distinctly upsample diverse content and downsample the redundant cluster to balance the distribution. The abundant cross-dataset evaluation demonstrates the impressive generalizability of the proposed SynDR-IQA.

**Questions:**

1. Can Figure 1 add some image samples? The Figure 1 only presents abstract discrete points, maybe providing a few corresponding image samples can make the figure more understandable, especially the images with high and low quality.
2. How the content upsampling actually works? Does it just select additional images with related content from another dataset (KADIS-700K)? In this way, can a larger pool of reference images (maybe from diverse large-scale datasets) be used for selection?
3. Is the content upsampling time-consuming? It seems selecting reference images from a large dataset can be hard to implement and time-consuming.

**Ethical Concerns:**

["NO or VERY MINOR ethics concerns only"]

**Final Justification:**

The authors have well addressed my concern using a convincing small-scale experiment. I will modify my score to accept.

**Limitations:**

yes

**Paper Formatting Concerns:**

No concerns

**Quality:**

3

**Strengths And Weaknesses:**

Strength:
1. The figure 1 well illustrates the limitation of training BIQA methods using synthetic data. The visualization on large-scale KADID-10K provided convincing evidence of the key challenge of distribution imbalance.
2. The paper writing is clear and easy to understand. The introduction gives a clear entry into the current challenge of data distribution of synthetic data and elicits the explicit solution to reshape the distribution.
3. Understandable theoretical analysis and extensive experimental results. The cross-dataset evaluation are based on multiple settings, including syn-to-real test.

Weakness:
1. Lack of comparison with more training data. The paper presents a potential solution to improve the generalizability of BIQA methods using distribution reshaping. However, directly using a larger training dataset can also solve the problem. The proposed SynDR-IQA is only effective if it can achieve similar performance compared to the same model trained on a much larger dataset, which means the SynDR-IQA can save a significant amount of native training data. In my opinion, an ablation experiment can be added that compares the SynDR-IQA and using a larger dataset to make the framework more convincing. Please note that it is not mandatory to complete the experiment during rebuttal since training with large dataset is very time-consuming. I recommend the authors can add this experiment in their final version.
2. Figure 1 lacks some image samples, making it somewhat abstract. The figure 2 contains some samples that make it more vivid.
3. No examples of content upsampling. Providing some original training set and corresponding selected reference image for upsampling can be very helpful to understand.

---

> ### Author Rebuttal · Authors · 2025-07-31
>
> Thank you very much for your detailed review and valuable feedback. Your recognition is a great encouragement for us to continue improving this work, and the questions you raised are crucial for enhancing the quality of our paper. We will address each of your points below.
>
> ---
>
> **Weakness 1: Lack of comparison with more training data**
>
> We appreciate this insightful comment. We completely agree that a comparison with models trained on larger datasets would make our framework more convincing.
>
> First, we would like to clarify that, as revealed by previous work (e.g., [1]), simply increasing samples for synthetic datasets does not always lead to improved generalization. The distribution gap between synthetic and real-world data might be a more critical bottleneck than the amount of data. Our work focuses precisely on this issue, aiming to enhance model generalization by reshaping the data distribution.
>
> Second, KADID-10k is currently the largest synthetic dataset with subjective annotations in the IQA field. This makes it difficult for us to find a larger, comparable dataset for a direct comparison.
>
> In response to your suggestion, we conducted a new experiment. We applied our method to TID2013 [2], a dataset much smaller than KADID-10k (with only 25 reference images). Since distortion generation code is not publicly available for synthetic datasets other than KADID-10k, we selected the distortion types common to both TID2013 and KADID-10k (#1, #8, #10, #11, #16, #17) and followed the distortion level settings of TID2013 to implement them.
> Ultimately, we used a subset of distorted images from TID2013 (750 images in total) to train our model and compared its performance against a baseline model (ImageNet pre-trained ResNet-50) trained on the full KADID-10k dataset (10,025 images):
>
> |Method|Train Dataset|Labeled Images|LIVEC (SRCC/PLCC)|KonIQ-10k (SRCC/PLCC)|
> |---|---|---|---|---|
> |**Baseline**|KADID-10k|10,025|0.565 / 0.581|0.625 / 0.613|
> |**SynDR-IQA**|TID2013 (Part)|750|**0.611 / 0.609**|**0.664 / 0.693**|
>
> The results clearly show that our SynDR-IQA method, using only approximately 7.5% of the baseline's data, significantly outperforms the baseline model on both the LIVEC and KonIQ-10k datasets. This experiment strongly demonstrates the value of our method.
>
> [1] Ponomarenko N, Ieremeiev O, Lukin V, et al. Color image database TID2013: Peculiarities and preliminary results[C]//European workshop on visual information processing (EUVIP). IEEE, 2013: 106-111.
> [2] Li A, Wu J, Liu Y, et al. Bridging the synthetic-to-authentic gap: Distortion-guided unsupervised domain adaptation for blind image quality assessment[C]//Proceedings of the IEEE/CVF Conference on Computer Vision and Pattern Recognition. 2024: 28422-28431.
>
> ---
>
> **Weakness 2 & Question 1: Figure 1 lacks image samples**
>
> Thank you for the valuable suggestion. Adding specific image samples to the UMAP distribution plot in Fig. 1 would make it more intuitive and easier to understand.
>
> However, due to the NIPS rebuttal policy, we are unable to submit an updated figure at this stage. We commit to adding several representative image samples (especially high-quality and low-quality samples corresponding to the data points) to Fig. 1 in the final version of the paper to enhance its readability.
>
> ---
>
> **Weakness 3: No examples of content upsampling**
>
> This is an excellent suggestion. In fact, Figure 2 already includes partial examples of content upsampling, illustrating different scenarios in candidate reference selection: cases where a candidate is too different from the training set references (top left), too similar (bottom right), and pairs of candidates that are too similar to each other (bottom left).
>
> To provide a clearer understanding, we will add more detailed examples in the appendix of the final paper, including:
> - Reference images from the original training set, with their corresponding distorted images and MOS values.
> - New reference images selected from the candidate set, along with their corresponding distorted images and generated pseudo-labels.
>
> Furthermore, along with the open-sourced code, we will provide all candidate set images and related information to facilitate reproduction and deeper understanding for other researchers.
>
> ---
>
> **Question 2: How content upsampling works**
>
> Thank you for your question regarding the details of our method. The process is as follows: We first use a baseline BIQA model (trained on KADID-10k) to filter relatively high-quality images from the KADIS-700k dataset. From these, we randomly sample a number of images equal to the number of reference images in KADID-10k to construct our candidate set for content upsampling. We then use an ImageNet pre-trained ResNet-50 to extract features from the reference images in the original training set and the images in the candidate set. Following Algorithm 1, we identify suitable new reference images from the candidate set and then generate new distorted samples and pseudo-labels for them using Algorithm 2.
>
> Regarding the question of "why not use a larger, more diverse pool of reference images," this is theoretically feasible, as a larger pool could introduce greater content diversity. However, in practice, this approach faces two main challenges that could introduce more noise and degrade model performance:
>
> 1. Large-scale image pools (like KADIS-700k) contain a significant number of low-quality images, which would lead to the generation of unreliable pseudo-labels.
> 2. The limited content diversity in KADID-10k's reference set constrains the ability of our nearest-neighbor-based strategy to generate accurate pseudo-labels.
>
> Our current design strikes a fine balance between enriching content diversity and avoiding excessive noise.
>
> Notably, as mentioned in our response to Weakness 1 of Reviewer 1mY6, we found that models pre-trained with CLIP might facilitate the generation of higher-quality pseudo-labels, offering a potential direction for more reliably leveraging large-scale candidate sets in the future.
>
> ---
>
> **Question 3: Is it time-consuming?**
>
> Thank you for the valuable suggestion. The process is highly efficient. Excluding the generation of distorted images (which are pre-generated), the entire content upsampling process takes only about 14.5 seconds. Moreover, for a given backbone and training set, content upsampling is a one-time preprocessing step. Considering that this  tiny extra computation yields a significant performance gain in model generalization, we believe it is a highly efficient and valuable component of our framework.
>
> ---
>
> Once again, thank you for your valuable time and constructive feedback. We hope our responses have adequately addressed your concerns.

---

> > ### Comment · Reviewer_BpnP · 2025-08-01
> >
> > Thanks the authors for their efforts. The authors used an example experiment to significantly address my concern, where the proposed method can outperform the same architecture that is trained with much more data. This means the proposed method can actually save the volume of training data, and maybe bridge the syn-to-real gap. Therefore, I am willing to lift my score to weak accept.

---

### Decision · Program_Chairs · 2025-09-17

**Decision:**

Accept (poster)

**Comment:**

This paper addresses the common synthetic-to-real generalization issue in BIQA by identifying a key observation: synthetic datasets tend to induce clustered and discrete feature distributions, which hinder model generalization. To solve this, the authors propose SynDR-IQA, a theoretically grounded data reshaping framework that combines content upsampling and redundant cluster downsampling to improve training data distribution. The method consistently improves cross-dataset performance.

While some reviewers were concerned about the novelty of the approach, the authors clarified that their primary contribution lies not in model architecture design but in providing a theoretically justified and data-centric perspective on the root causes of generalization failure in BIQA. This shift in focus opens new research directions for the field. Additionally, they provided new experiments, including outperforming baselines with less training data and leveraging CLIP encoders to enhance content selection, that demonstrate the method’s practical effectiveness and extensibility. Most reviewers responded positively following the rebuttal. Considering the importance of the problem and the novelty of the analysis perspective, the AC recommends accepting this paper.